# Understanding Adam through the Lens of Duality: A Unified Theory of Normalized Gradient Methods

## Abstract

This paper presents a fresh mathematical perspective on Adam, whose empirical success is in stark contrast with its analytic intractability. We derive Adam via duality, showing that many of its design choices such as coordinate-wise normalization and exponential moving averages emerge naturally from a unified framework. Using this framework, we first analyze two normalized gradient descent methods in the setting of linearly separable data which favor different solutions with differing geometries: SignGD, which converges to a $\ell_\infty$-max-margin classifier at a rate of $\mathcal{O}(\frac{1}{\sqrt{t}})$, and *Normalized GD*, which instead converges to a $\ell_2$-max-margin classifier at a rate of $\mathcal{O}(\frac{1}{t})$, vastly improving upon the $\mathcal{O}(\frac{1}{\ln t})$ rate for gradient descent. Next, we show that Adam, which replaces the solitary gradients within SignGD with exponential moving averages, achieves margin maximization at a rate of $\mathcal{O}(\frac{1}{\sqrt{t}})$, whereas prior work requires additional assumptions and has a rate of $\mathcal{O}(\frac{1}{t^{1/3}})$. In the stochastic setting, this duality approach gives the first high probability convergence guarantee for low test error with standard empirical choices of the momentum factors $0 < \beta_1 < \beta_2 < 1$, improving upon prior work which can only establish bounds in expectation, and has a slower rate of $\mathcal{O}(\frac{1}{t^{1/4}})$.

## 1 Introduction

First-order gradient methods are the standard mechanism used in machine learning to fit model parameters to data. This is primarily due to convenience of implementation thanks to modern auto-differentiation, and due to statistical problems inherently being resistant to sophisticated high-accuracy approaches such as Newton methods (Bottou, 2010). The Adam optimization method (Kingma & Ba, 2015) is the dominant choice for many modern machine learning applications (Groeneveld et al., 2024; Brown et al., 2020; Touvron et al., 2023; Ramesh et al., 2022; Polu & Sutskever, 2020; Ellis et al., 2024), and is given by the coordinate-wise recursion

$$(w_{s+1})_j := (w_s)_j - \eta_s \frac{\frac{1-\beta_1}{1-\beta_1^{s+1}} \sum_{k=0}^s \beta_1^{s-k} \nabla \widehat{\mathcal{R}}_k(w_k)_j}{\sqrt{\frac{1-\beta_2}{1-\beta_2^{s+1}} \sum_{k=0}^s \beta_2^{s-k} \nabla \widehat{\mathcal{R}}_k(w_k)_j^2} + \epsilon}, \qquad (1)$$

where $\eta_s$ is a *step size*, $w_s$ are parameters of the model, $\widehat{\mathcal{R}}_s(w_s)$ is the average loss over the $s^{\text{th}}$ minibatch, $\beta_1, \beta_2 \in (0, 1)$ are exponential weighting constants, $\epsilon$ is the stability constant. The goal of this paper is to understand how each component plays a role in the training dynamics and generalization properties of Adam; the coordinate-wise normalization and $\epsilon$ completely determine the implicit bias of Adam while the momentum factors $\beta_1$ and $\beta_2$ reduce stochastic noise.

In many machine learning tasks, the number of the model parameters typically exceed the number of training examples. As a consequence of overparameterization, the training objective generally has infinitely many solutions. Therefore, characterizing the set of solutions an optimization algorithm converges toward, or its implicit bias, sheds valuable insight on the optimization algorithm. In this vein, a line of work has investigated the implicit bias of Adam and other adaptive algorithms (Xie & Li, 2024; Wang et al., 2021; Fan et al., 2025; Qian & Qian, 2019). Specifically, (Xie & Li, 2024) showed that Adam converges to the $\ell_\infty$-max margin solution. Unfortunately, it is unclear from

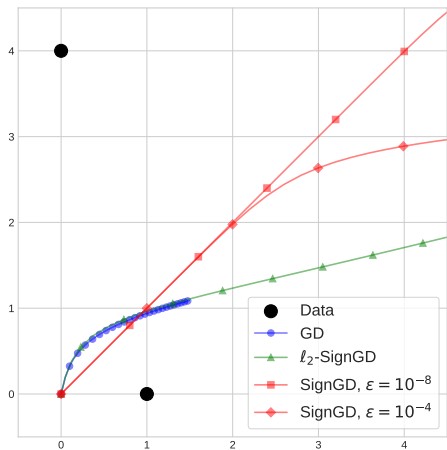 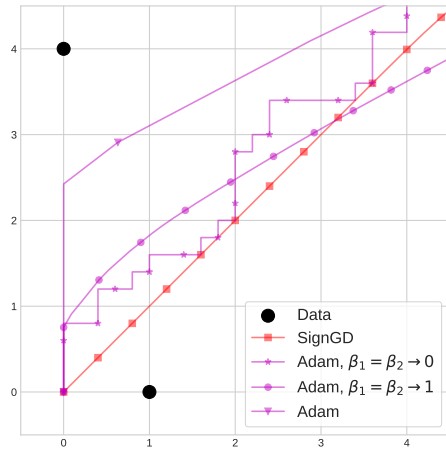

(a) GD with per-coordinate normalization (SignGD) or whole-gradient normalization (Normalized GD).

(b) SignGD (using full-batch gradients), and Adam (using stochastic gradients) with various $(\beta_1, \beta_2)$.

Figure 1: Figures 1a and 1b depict trajectories of various first-order methods on simple but illustrative data. In detail, the data consists of two points in $\mathbb{R}^2$, namely $(0, 4)$ and $(1, 0)$ with positive labels, corresponding to $\ell_\infty$-maximum-margin direction $u_\infty := (1, 1)$ and $\ell_2$-maximum-margin direction $u_2 := (4/\sqrt{17}, 1/\sqrt{17})$. GD and Normalized GD are seen to converge to $u_2$ in Figure 1a, albeit GD is very slow. SignGD exactly follows the diagonal direction $u_\infty$ until the norm of the loss gradient becomes similar in magnitude to the stability constant $\epsilon$, after which $\epsilon = 10^{-4}$ dominates the denominator, changing SignGD's implicit bias from maximizing the $\ell_\infty$-margin to maximizing the $\ell_2$-margin. Meanwhile Figure 1b shows SignGD and Adam, all ostensibly following $u_\infty$, though with significant differences. The stochastic methods (all labeled "Adam") use identical orderings of the examples, and thus the initial updates are all vertical by the same coincidence, but the length of this initial vertical segment varies.

prior work why Adam exhibits a $\ell_\infty$-max margin bias. To resolve this issue, we develop a duality framework which explains how this implicit bias arises as well as yielding faster margin maximization rates and new generalization guarantees.

**Contribution.** This work provides another approach to deriving Adam, SignGD, and Normalized GD. First we derive per-coordinate normalized gradient methods directly via a duality approach, with unified mirror descent analyses. Then the bounds on the moving averages are estimated separately to form the full analysis of Adam. To handle flexible choices of $\beta_1$ and $\beta_2$, some problem-dependence is necessary. Indeed, (Reddi et al., 2019) showed that for every choice of $(\beta_1, \beta_2)$, there exists an online convex optimization problem such that Adam can fail to converge. Hence, like (Zhang et al., 2024), the present work considers binary classification with linear predictors and logistic loss. Moreover, the analysis operates in the setting of *linear separability*, meaning it will be possible to pick a single $w$ with zero classification error (see Section 2 for details). Our main contributions can be summarized as follows.

- We develop a duality framework that shows exactly why Adam and SignGD are implicitly biased toward $\ell_\infty$-max margin solutions. Specifically, this arises as a consequence of Adam and SignGD minimizing a $\ell_1$-norm objective in the *dual*.

- We prove margin maximization rates for Adam, SignGD, and Normalized GD; notably, for Adam, we sharpen margin maximization rates from $\mathcal{O}(1/t^{1/3})$ to $\mathcal{O}(\frac{1}{\sqrt{t}})$.

- We show that the test error of stochastic mini-batch Adam is at most $\mathcal{O}(\frac{n}{\sqrt{d}})$, where $n$ is the number of samples. We additionally provide a matching lower bound $\Omega(d)$ for the sample complexity of Adam.

We further elaborate and contextualize our contributions below.

1. **Duality Framework (cf. Section 3).** In this section, we develop the duality framework. Given Hölder conjugates $p, q$, we first show that Adam and other normalized gradient methods are solving a $\ell_q$-minimization problem in the dual. Then, in Lemma 20, we establish an equivalence between the dual $\ell_q$-minimization problem and the primal $\ell_p$-margin maximization problem. In short, Adam is implicitly biased toward $\ell_\infty$-max margin solutions since it solves a $\ell_1$-minimization problem in the dual. This duality framework builds off the framework presented in Ji & Telgarsky (2019b). As technical contributions, we generalize the framework to handle stochastic updates (see Section 5) via a *ghost sample* technique and general $\ell_q$-dual objectives.

2. **Margin maximization for full-batch normalized methods (cf. Section 4).** This section considers the setting of full-batch gradients and linear separability. Here we show various margin maximization results for three algorithms; SignGD, Normalized GD, and Adam. Concretely, the margin of a classifer $w$ over a training set $\{(x_i, y_i)\}_{i \in [n]}$ is defined as $\min_i \left\langle \frac{w}{\|w\|}, y_i x_i \right\rangle$ where $\|\cdot\|$ is some norm. In Theorem 2, we show that SignGD and Normalized GD maximize the $\ell_\infty$- and the $\ell_2$-margin with rates $\mathcal{O}(\frac{1}{\sqrt{t}})$ and $\mathcal{O}(\frac{1}{t})$ respectively. Next, in Theorem 3, we prove Adam maximizes the $\ell_\infty$-margin at a rate of $\mathcal{O}(\frac{1}{\sqrt{t}})$, improving over the previous $\mathcal{O}(1/t^{1/3})$ margin maximization rate Xie & Li (2024), when the stability constant $\epsilon$ is 0. When the stability constant $\epsilon$ is nonzero, Theorem 3 shows the same $\ell_\infty$-margin maximization rate $\mathcal{O}(\frac{1}{\sqrt{t}})$ holds until the gradient norm $\left\|\nabla \widehat{\mathcal{R}}(w_s)\right\|_1$ falls below $\mathcal{O}(\sqrt{\epsilon})$; such a result cannot be strengthened to hold for all time as Adam with $\epsilon > 0$ asymptotically converges to the $\ell_2$-maximum margin solution (Wang et al., 2021). Indeed, this phenomenon can be observed in Figure 1b where various instantiations of Adam initially follow the $\ell_\infty$-max-margin direction and then switches to the $\ell_2$-max-margin direction. It is also notable that GD, without normalization, maximizes margins at a rate of $\mathcal{O}(1/\ln(t))$ or $\mathcal{O}(1/\sqrt{\ln(t)})$, depending on details of the setting (Soudry et al., 2017; Ji & Telgarsky, 2018).

3. **Population loss minimization with high probability with minibatches (cf. Section 5).** In this section, we consider Adam with i.i.d stochastic minibatches. In particular, Theorem 5 analyzes minibatch Adam (with $0 < \beta_1 \leq \beta_2 < 1$), showing that the test loss is at most $\mathcal{O}(\frac{d}{\sqrt{n}})$ with high probability, where $n$ is the number of samples. The dimension factor $d$ in the test error bound is unavoidable as demonstrated by Theorem 6 which constructs a data distribution where the ($\ell_\infty$) margin is constant yet Adam still requires $\Omega(d)$ samples to achieve low test error. The proof of Theorem 5 combines the aforementioned duality techniques and a modified perceptron proof adapted toward dual variables. To the best of our knowledge, this is the first high probability test error guarantee, under typical $\beta_1, \beta_2$ values. The closest analysis in the literature has a rate which translates to $\mathcal{O}(\sqrt{d}/n^{1/4})$ in this classification setting, and is proved in expectation (Défossez et al., 2022).

The rest of this paper is organized as follows. The following section closes this introduction with brief notation and assumptions, Section 3 develops the duality framework as well as useful mirror descent guarantees, Section 4 gives the full-batch margin maximization analysis, Section 5 gives the stochastic Adam analysis, Section 6 provides further related work, and Section 7 closes with open problems and future directions.

## 2 PRELIMINARIES

**Notation and Assumptions.** Norms are disambiguated with a subscript, as $\|\cdot\|_1$, $\|\cdot\|_2$, and $\|\cdot\|_\infty$ all appear frequently; in a few places, the analysis applies to arbitrary norms, and simply $\|\cdot\|$ is written. Furthermore, $\langle \cdot, \cdot \rangle$ denotes inner product.

Data $((x_i, y_i))_{i=1}^n$ has $x_i \in \mathbb{R}^d$ and $y_i \in \{-1, 1\}$, with $\|x_i\|_* \leq 1$ in the appropriate dual norm. Inputs and their labels are collected into single vectors $z_i := y_i x_i$, which are then collected as rows $(z_i^\top)_{i=1}^n$ of a matrix $Z \in \mathbb{R}^{n \times d}$. Correspondingly, $Zw_t$ is the unnormalized margin on all data points at time $t$. The loss, as above, is always the logistic loss $\ell(r) = \ln(1 + \exp(-r))$.

Throughout, standard concepts from convex analysis will be used, e.g., $f^*(s) = \sup_x \left( \langle s, x \rangle - f(x) \right)$ will denote the *Fenchel conjugate* of a convex function $f$; for more on convexity and the key role of this function, see for instance (Hiriart-Urruty & Lemaréchal, 2001; Borwein & Lewis, 2000).

The batch methods will use the following notion of batch margin.

**Assumption 1.** *Suppose the data is* linearly separable, *meaning there exists* $u \in \mathbb{R}^d$ *with* $Zu > 0$ *(coordinate-wise) and define margins*

$$\gamma_2 := \max_{\|u\|_2 \leq 1} \min_i z_i^\mathsf{T} u, \qquad \gamma_\infty := \max_{\|u\|_\infty \leq 1} \min_i z_i^\mathsf{T} u,$$

*respectively for Normalized GD, SignGD, and Adam; context will disambiguate, thus simply* $\gamma$ *is often written to declutter. (Linear separability means* $\gamma_2 > 0$ *and* $\gamma_\infty > 0$.*)*

The stochastic case is similar, with some care for measure-theoretic issues.

**Assumption 2.** *Given a distribution* $\mathcal{D}$ *over* $z = yx$, *let* $\mathrm{supp}(\mathcal{D})$ *denote its support. Define corresponding margin notions as*

$$\gamma_2 := \sup_{\|u\|_2 \leq 1} \inf_{z \in \mathrm{supp}(\mathcal{D})} \langle z, u \rangle, \qquad \gamma_\infty := \sup_{\|u\|_\infty \leq 1} \inf_{z \in \mathrm{supp}(\mathcal{D})} \langle z, u \rangle.$$

*The assumption is that* $\mathcal{D}$ *is linearly separable, which means* $\gamma_2 > 0$ *and* $\gamma_\infty > 0$.

# 3 DUALITY FRAMEWORK AND MIRROR DESCENT.

**Dual analysis.** We present the framework for deriving various normalized gradient methods including SignGD, $\ell_2$ normalized GD, Adam. We first develop some additional notation. Define $\mathcal{L}(v) = \frac{1}{n} \sum_{i=1}^n \ell(v_i)$, whereby $\mathcal{L}(Zw) = \widehat{\mathcal{R}}(w)$, and define the *smooth margin* $\psi(v) = \ell^{-1}(n\mathcal{L}(v))$. Note that for logistic loss, we have the following inequality

$$\psi(Zw) \leq \min_{i \in [n]} \langle w, z_i \rangle.$$

Hence, it suffices to lower bound $\psi(Zw)$ to obtain a unnormalized margin bound for weight $w$. As $\psi$ is smooth and concave, we will work entirely with $\psi$.

We start by considering the margin maximization problem

$$\max_{\|w\| \leq 1} \min_{i \in [n]} \langle w, y_i x_i \rangle,$$

which, by Lemma 20, has the corresponding dual problem

$$\min_{q \in \Delta : \psi^*(q) \leq 0} \|Z^\mathsf{T} q\|_* . \tag{2}$$

Now given an arbitrary sequence $\{g_s\}_{s \leq t} \subset \mathbb{R}^d$ and a learning rate $\eta > 0$, consider the generic update rule for the weights $w_s$,

$$w_{s+1} = w_s - \eta g_s.$$

For the remainder of the paper, assume $w_0 = 0$ for all gradient methods. Let $q_s := -\nabla \psi(Zw_s)$. In the case of GD, $g_s = \nabla \widehat{\mathcal{R}}(w_s)$, and Ji & Telgarsky (2019b) made the crucial observation that the induced update rule for $q_s$ coincided with the following mirror descent update,

$$\nabla(-\psi)^*(q_{s+1}) = \nabla(-\psi)^*(q_s) - \nabla_q f(q_s),$$

where $f(q) := \frac{1}{2}\|Z^\mathsf{T} q\|_2^2$. In other words, the (dual) variables $q_s$ were explicitly solving the dual problem eq. (2). This connection makes it extremely clear why GD converges to the $\ell_2$ max margin solution. Notice how we started with a primal update for $w_s$ and then derived the induced update for $q_s$. We can also do the reverse. Namely, by varying the dual objective $f(q)$, we can derive new primal algorithms with different implicit biases.

Set $f(q) = \|Z^\mathsf{T} q\|_1$, and let the dual variables $q_s$ be updated as follows

$$\nabla(-\psi)^*(q_{s+1}) = \nabla(-\psi)^*(q_s) - \nabla_q f(q_s).$$

As $-\psi$ is strongly convex, it follows that $Zw_s = \nabla(-\psi)^*(q_s)$ and hence the induced primal update is

$$Zw_{s+1} = Zw_s - \eta Z \, \mathrm{sign}(Z^\mathsf{T} q_s).$$

Since $Z^\mathsf{T} q_s$ is $-\nabla_w \psi(Zw_s)$, we have that $\text{sign}(Z^\mathsf{T} q_s)$ is $\text{sign}(\nabla \widehat{\mathcal{R}}(w_s))$. Therefore, by letting $w_{s+1}$ be any update satisfying the preceding equation, we have the following iterative update for the primal variables,

$$w_{s+1} = w_s - \eta\,\text{sign}(\nabla \widehat{\mathcal{R}}(w_s)), \tag{3}$$

which is exactly SignGD. As the dual objective is $f(q) = \|Z^\mathsf{T} q\|_1$, by Lemma 20, it is immediate that SignGD is implicitly solving the $\ell_\infty$-margin-maximization problem. In a similar fashion, setting $f(q) = \|Z^\mathsf{T} q\|_2$ gives rise to Normalized GD,

$$w_{s+1} = w_s - \eta\,\frac{\nabla \widehat{\mathcal{R}}(w_s)}{\left\|\nabla \widehat{\mathcal{R}}(w_s)\right\|_2}. \tag{4}$$

**Mirror descent.** Next we provide a nonstandard mirror descent guarantee which will be used to analyze the dual variables $q_s$. Let the Bregman divergence be defined as

$$D_h(u, u_s) := h(u) - h(u_s) - \langle v_s, u - u_s \rangle,$$

where $u_s = \nabla h^*(v_s)$.

**Lemma 1.** *Given initial iterates* $u_0, v_0 \in \mathbb{R}^d$, *learning schedule* $\{\eta_s\}_{s<t} \subset \mathbb{R}$, *and a sequence* $\{\xi_s\} \subset \mathbb{R}^d$, *suppose the iterates* $v_s$ *follow the update rule,*

$$v_{s+1} = v_s - \eta_s \xi_s.$$

*Let $h$ be any closed, proper, convex function and let $h^*$ be the convex conjugate of $h$. Suppose* $u_s = \nabla h^*(v_s)$. *Then, the sequence* $\{u_s\}_{s<t}$ *can equivalently be generated by mirror descent,*

$$u_{s+1} = \arg\min\left\{\eta_s \langle \xi_s, u \rangle + D_h(u, u_s) \,:\, u \in \mathbb{R}^d\right\}.$$

*In addition, if $h$ is $\lambda_s^{-1}$ strongly convex over the line segment $[u_s, u_{s+1}]$ with respect to $\|\cdot\|$,*

$$h^*(v_0) - h^*(v_t) \geq \sum_{s<t} \eta_s \langle \xi_s, u_s \rangle - \frac{\eta_s^2 \lambda_s}{2}\|\xi_s\|_*^2.$$

Of note, the preceding lemma considers generic updates $\xi_s$ and hence can be specialized to handle various normalized gradient methods such as SignGD and Adam. In addition, this lemma will be used to prove both margin maximization and population loss minimization though with different choices of mirror potential $h$. Most importantly, the mirror descent analysis of the dual variables provides a control on the *primal gap* $h^*(v_0) - h^*(v_t)$. We make a slight technical remark on the appearance of the strong convexity constant as its reciprocal in the bound of Lemma 1 is common in $\mathcal{O}(\frac{1}{\sqrt{t}})$-type mirror descent analyses (Bubeck, 2015) and comes from lower bounding $\langle \eta_s \xi_s, u_{s+1} - u_s \rangle + D_h(u_{s+1}, u_s)$ via the standard application of strong convexity of $h$ and Fenchel-Young inequality.

## 4 MARGIN MAXIMIZATION WITH FULL-BATCH METHODS

In this section, we will present margin maximization results for various normalized gradient methods. Concretely, Theorem 2 show SignGD maximizes the $\ell_\infty$-max-margin at a rate of $\mathcal{O}(\frac{1}{\sqrt{t}})$ and Normalized GD maximizes the $\ell_2$-max-margin at a faster rate of $\mathcal{O}(\frac{1}{t})$. Finally, Theorem 3 will show Adam maximizes the $\ell_\infty$-margin at a rate of $\mathcal{O}(\frac{1}{\sqrt{t}})$.

**Theorem 2.** *Suppose Assumption 1 holds. Take $\|\cdot\|$ to be $\|\cdot\|_\infty$ and $\|\cdot\|_2$ for SignGD and Normalized GD respectively. Assume that for all $i \in [n]$, the data is bounded, $\|x_i\|_* \leq 1$. Suppose the iterates $w_s$ are updated via SignGD, meaning eq. (3), with learning rate $\eta = \frac{\gamma_\infty}{4\sqrt{t}}$. Then for $t \geq \frac{8n^2}{\gamma_\infty^4}$, SignGD maximizes the $\ell_\infty$-margin:*

$$\frac{\psi(Zw_t)}{\|w_t\|_\infty} \geq \gamma_\infty - \frac{8n}{\gamma_\infty \sqrt{t}}.$$

*If the iterates $w_s$ are updated using Normalized GD, meaning eq. (4), with learning rate $\eta = \frac{\gamma_2}{4}$, then for $t \geq \frac{ln^2(4n)}{\gamma^2}$, Normalized GD maximizes the $\ell_2$-margin:*

$$\frac{\psi(Zw_t)}{\|w_t\|_2} \geq \gamma_2 - \frac{16\ln(4n)}{\gamma_2 t}.$$

Before sketching the proof, we first make several remarks.

1. Note that Normalized GD enjoys a faster margin maximization rate of $\mathcal{O}(\frac{1}{t})$ despite the fact that the corresponding dual objective $f(q) = \|Z^\mathsf{T} q\|_2$ is nonsmooth. This is because an alternative way to view Normalized GD is that the dual variables are minimizing a smooth objective $f(q) = \frac{1}{2}\|Z^\mathsf{T} q\|_2^2$ with an adaptive learning rate $\eta_s = \frac{\eta}{\|Z^\mathsf{T} q_s\|}$.

2. On the other hand, the same trick fails for SignGD as both $\|Z^\mathsf{T} q\|_1$ and its square $\|Z^\mathsf{T} q\|_1^2$ are nonsmooth which results in a slower rate of $\mathcal{O}(\frac{1}{\sqrt{t}})$.

We provide a proof sketch of SignGD and defer the remaining details to the appendix. The proof of $\ell_2$ is slightly more involved to get both $\mathcal{O}(\frac{\ln(n)}{t})$ rate. We first invoke Lemma 1 with the following instantiation: mirror potential $h = (-\psi)^*$, primal iterates $v_s = Zw_s$, dual iterates $u_s = q_s$, primal update $g_s = \text{sign}(\nabla\widehat{\mathcal{R}}(w_s))$, dual update $\xi_s = Zg_s$, and learning rate $\eta = \frac{\gamma_\infty}{4\sqrt{t}}$.

Hence, by Lemma 1 and since by Lemma 20 the dual objective satisfies $\|Z^\mathsf{T} q_s\|_1 \geq \gamma_\infty$,

$$\psi(Zw_t) \geq \sum_{s<t} \eta \left[ \langle Zg_s, q_s \rangle - \frac{\eta \|Z^\mathsf{T} q_s\|_1}{2\gamma} \|Zg_s\|_\infty^2 \right] + \psi(Zw_0)$$

$$\geq \sum_{s<t} \eta \left( 1 - \frac{1}{8\sqrt{t}} \right) \|Z^\mathsf{T} q_s\|_1 + \psi(Zw_0)$$

$$\geq \sum_{s<t} \eta \left( 1 - \frac{1}{8\sqrt{t}} \right) \gamma_\infty + \psi(Zw_0). \tag{5}$$

As $w_0 = 0$, the initial smoothed margin satisfies $\psi(Zw_0) = -\ell^{-1}(n\ln(2)) \geq -n$. By the choice of the learning rate and since $t \geq \frac{8n^2}{\gamma_\infty^4}$, the right hand side of eq. (5) is nonnegative.

To finish the proof from here, it suffices to divide $\|w_t\|_\infty$ across both sides of eq. (5), and upper bound $\|w_t\|_\infty$. By a simple application of triangle inequality and noting that each entry of the SignGD update is bounded by 1, we obtain that $\|w_t\|_\infty \leq \sum_{s<t} \eta = \eta t$.

Putting it altogether,

$$\frac{\psi(Zw_t)}{\|w_t\|_\infty} \geq \frac{1}{\|w_t\|_\infty} \left( \sum_{s<t} \eta \left( 1 - \frac{1}{8\sqrt{t}} \right) \gamma_\infty - n \right)$$

$$\geq \gamma_\infty - \frac{8n}{\gamma_\infty \sqrt{t}}.$$

**Adam.** In this section, we prove various margin maximization results for Adam with full batch gradients. As indicated by Figure 1, the stability constant $\epsilon$ plays a nontrivial role of determining the implicit bias of Adam. Indeed, if $\epsilon > 0$, Adam asymptotically becomes GD and converges to the $\ell_2$-max-margin classifier which has been shown in Wang et al. (2021). This effect becomes more apparent as the norm of the gradient $\left\|\nabla\widehat{\mathcal{R}}(w_s)\right\|$ shrinks to the same magnitude as $\epsilon$. On the flip side, if $\epsilon = 0$, Adam is biased toward $\ell_\infty$-max-margin solutions for all time.

**Theorem 3.** *Suppose Assumption 1 holds with $\gamma = \gamma_\infty$ and $\|x_i\|_1 \leq 1$ for every $i \in [n]$. Let $C := \sqrt{\frac{1-\beta_1}{1-\beta_2}}$. For any $0 \leq \beta_1 \leq \beta_2 < 1$ and $0 < \epsilon \leq 1$, for constant learning rate $\eta = \frac{(1-\sqrt{\beta_2})^2 \gamma}{24C^2} \frac{1}{\sqrt{t}}$ where Adam iterates are updated using eq. (1), for all time $t$ such that $\left\|\nabla\widehat{\mathcal{R}}(w_t)\right\|_1 \geq d\sqrt{\epsilon}$, Adam maximizes the $\ell_\infty$ margin,*

$$\frac{\psi(Zw_t)}{\|w_t\|_\infty} \geq \gamma \left( 1 - \sqrt{\epsilon} \right) \sqrt{\frac{1-\beta_2}{1-\beta_1}} - \frac{24C^3}{\gamma \left( 1 - \sqrt{\beta_2} \right)^2 \sqrt{t}}. \tag{6}$$

Several remarks are in order. If $\epsilon = 0$, the theorem holds for all time. Hence, if $\beta_1 = \beta_2$, Adam converges to $\ell_\infty$ max margin classifer at a rate $\mathcal{O}(1/\sqrt{t})$. As the proof relies on arguing that the Adam update is approximately SignGD, it inherits the $\mathcal{O}(1/\sqrt{t})$ rate from the SignGD proof which is superior to $\mathcal{O}(1/t^{1/3})$ rate given by Zhang et al. (2024). It should be noted that Zhang et al. (2024) converges to the true margin $\gamma_\infty$ instead of $\sqrt{\frac{1-\beta_2}{1-\beta_1}}\gamma_\infty$. However, they rely on an additional assumption that each coordinate of the gradient is bounded away from zero by a positive constant $\rho$. By making the same assumption, we can likewise remove the $\sqrt{\frac{1-\beta_2}{1-\beta_1}}$ factor.

In the remainder of the section, we will provide a proof sketch of this theorem. For simplicity, we only consider the case where $\epsilon = 0$. As with the SignGD proof, we apply Lemma 1 with the following instantiation: the mirror potential $h = (-\psi)^*$, primal iterates $v_s = Zw_s$, dual iterates $u_s = q_s$, primal update $g_s$ as defined in eq. (1), dual update $\xi_s = Zg_s$, and learning rate $\eta$ as given in Theorem 3. Therefore, Lemma 1 gives

$$\psi(Zw_t) - \psi(Zw_0) \geq \sum_{s=0}^{t} \left[ \eta \langle Zg_s, q_s \rangle - \frac{\eta^2}{2} \|Zg_s\|_\infty^2 \right].$$

Abbreviating $\sigma_s := \text{sign}(\widehat{\mathcal{R}}(w_s))$, we have that the inner product term $\langle Zg_s, q_s \rangle$ can be decomposed,

$$\langle Zg_s, q_s \rangle = \langle Z\sigma_s, q_s \rangle - \langle Zg_s - Z\sigma_s, q_s \rangle = \|Z^\intercal q_s\|_1 - \langle Zg_s - Z\sigma_s, q_s \rangle.$$

It suffices to bound the second term as the first term can be handled in the same manner as in the SignGD proof. The following lemma controls the second term, namely the deviation between Adam and SignGD updates.

**Lemma 4** (Informal version of Lemma 18). *Fix $\epsilon \geq 0$, then*

$$\left| \left\langle \sigma_s - g_s, \nabla \widehat{\mathcal{R}}(w_s) \right\rangle \right| \leq \frac{12C^2}{\left(1 - \sqrt{\beta_2}\right)^2} \eta \widehat{\mathcal{G}}(w_s),$$

*where $\widehat{\mathcal{G}}(w) = -\frac{1}{n}\sum_{i=1}^{n} \ell'(\langle w, z_i \rangle)$.*

As in proof of Lemma A.3 in (Zhang et al., 2024), the proof proceeds by showing the moving average of the first and second moment of the gradients are close to the current gradient. We make several refinements to improve the dependency on dimension $d$ and $\eta$, the latter being critical for obtaining a faster margin maximization rate. To be more explicit, we first discuss the improved dependence on $\eta$. The proof of Lemma 4 relies on Lemma 16 which argues that the averaged first moment $m_s$ and the averaged second moment $v_s$ of the gradients are close to the current gradient and squared gradient respectively. The improved dependence on $\eta$ comes from a better control on the deviation between $\sqrt{v_s}$ and $\left|\nabla\widehat{\mathcal{R}}(w_s)\right|$. The improved dependence on dimension $d$ comes from bounding the deviation instead of worse case bounding the deviation of each coordinate. With Lemma 4, we can prove Theorem 3 by applying the same proof strategy used to show Theorem 2.

## 5 POPULATION LOSS MINIMIZATION WITH STOCHASTIC GRADIENTS

In this section, we consider mini-batch Adam in the online setting; in particular, we use the update in eq. (1), with a minibatch $S_s$ of size $B = |S_s|$. For convenience, the update will be written as

$$w_{s+1} = w_s - \eta_s g_s.$$

Here, $g_s$ collects the various terms of eq. (1):

$$g_{sj} = \frac{\sum_{s \leq t} a_{t,s} \nabla\widehat{\mathcal{R}}_s(w_s)_j}{\sqrt{\sum_{s \leq t} b_{t,s} \nabla\widehat{\mathcal{R}}_s(w_s)_j^2 + \epsilon}},$$

where the numerator and denominator weightings and their rescaling are packaged within $a_{t,s}$ and $b_{t,s}$. Under this setting, Theorem 5 shows that Adam with mini-batch size of $\mathcal{O}(\sqrt{n})$ and iteration count $t = \sqrt{n}$ will achieve low test error $\mathcal{O}(\frac{d}{\sqrt{n}})$. Furthermore, Theorem 6 provides a lower bound on the sample complexity of Adam, showing that the dimension factor $d$ in the test error bound of Theorem 5 is unavoidable.

**Theorem 5.** *Assume the distributional linear separability: Assumption 2 holds with $\gamma = \gamma_2$. Further assume, for $z = yx \sim \mathcal{D}$, the data satisfies $\|z\|_2 \leq 1$ and $\|z\|_1 \leq R$ almost surely. For any $0 \leq \beta_1 \leq \beta_2 < 1$ and $0 < \epsilon \leq 1$, for constant learning rate $\eta = \frac{\epsilon^2(1-\beta_1)^{5/2}\gamma_2}{4R^2}$, batch size $B = \sqrt{n}$, and iteration count $t = \sqrt{n}$, with probability at least $1 - 10\delta$, Adam achieves low test error:*

$$\min_{s<t} \Pr_z(\langle w_s, z \rangle < 0) \leq \frac{2048R^2}{(1-\beta_1)^{5/2}\epsilon^2\gamma_2^3\sqrt{n}}. \tag{7}$$

A more detailed comparison to the literature is as follows. The only other analysis of Adam with stochastic gradients is (Défossez et al., 2022), which gives a rate of $\mathcal{O}(1/\sqrt{n})$, after trading off various choices in their Theorem 4, in terms of squared gradient norms. In our setting, our proof is ultimately controlling something on the same order as the unsquared gradient norm (namely, the derivative of the loss), and thus the apples-to-apples rate is $\mathcal{O}(1/n^{1/4})$. Moreover that rate is in expectation, which is important as the denominator in Adam makes deviations and high probability harder to control.

All that said, the bound here has two deficiencies. One is the dimension dependence. Unfortunately, while the dimension factor $d$ in the test error guarantee is unsavory, it is unavoidable. Indeed, Theorem 6 provides a data distribution which is separable with a dimension-free margin, yet the sample complexity of Adam is $\Omega(d)$. A second issue is that the rate for Adam is worse in terms of constants; there are many technical hurdles to overcome to make the rate better, and we leave this interesting question to future work as well.

The proof superficially invokes a perceptron-style argument, but features a number of innovations; the first of these is to track progress on a *ghost sample*. Fix $N > 0$ and sample another $N$ data points $\{\overline{x}_i, \overline{y}_i\}_{i \in [N]}$ iid from the same population distribution. Note that the algorithm *never* sees this dataset and hence we can make $N$ arbitrarily large. Let $\overline{Z}$ be the matrix such that it's $i$-th row is $\overline{z}_i^\top := \overline{y}_i \overline{x}_i^\top$. Denote

$$\overline{G}(w) := \frac{1}{N} \sum_{i \in [N]} -\ell'(\langle w, \overline{z}_i \rangle), \quad G(w) := \mathbb{E}\left[-\ell'(\langle w, z \rangle)\right].$$

The function $G$ is an upper bound on the test zero-one error. Namely,

$$\Pr_z(\langle w, z \rangle < 0) \leq \frac{1}{\ln^2(2)} G(w)^2.$$

Hence, to get a good test error it suffices to control $G(w)$. Now let $\overline{\mathcal{R}} : \mathbb{R}^N \to \mathbb{R}$ to be the function such that

$$\overline{\mathcal{R}}(\xi) = \frac{1}{N} \sum_i \ell(\xi_i),$$

and define $q_s := \nabla \overline{\mathcal{R}}(\overline{Z}w_s)$, $p_s := \overline{Z}w_s$.

**Primal and Dual update.** Define $h = \overline{\mathcal{R}}^*$. By contrast to Section 4 and prior work (Ji & Telgarsky, 2019b), here the duality is formed with loss directly without any normalization. Then, by convexity, $p_s = \nabla h(q_s)$. Otherwise, proceeding is in the earlier dual derivation, starting from the Adam update equation 1 and multiplying $\overline{Z}$ on both sides grants,

$$p_{s+1} = p_s - \eta_s \overline{Z} g_s,$$

which is equivalent to

$$\nabla h(q_{s+1}) = \nabla h(q_s) - \eta_s \overline{Z} g_s,$$

which is the mirror descent update rule for the dual variables $q_s$.

With this key concept in hand, the proof proceeds via a similar analysis to Lemma 1, but then using standard perceptron steps to introduce the margin and bound the predictor norm.

The following theorem shows the existence of a data distribution that is separable with constant margin, yet the sample complexity is $\Omega(d)$ for Adam.

**Theorem 6.** *Suppose iterates are updated via eq. (1) where $g_s$ can be a stochastic or full batch update. Let $n$ denote the total number of data samples. There exists a data distribution such that if $\{(x_i, y_i)\}_{i=1}^n \sim \mathcal{D}$ and if $n \leq \frac{d}{2}$, then for all time $s \leq t$, the misclassification error is nonvanishing,*

$$\Pr_{(x,y)}(y \langle w_s, x \rangle < 0) \geq \frac{1}{2}.$$

The proof is by construction. In particular, consider the uniform distribution over the standard basis vectors in $\mathbb{R}^d$ with all positive labels. By inspection, one notes that the loss gradients are a linear combination of the sampled data. Hence, if the number of data points sampled is less than $\frac{d}{2}$, for any iterate $w_s$, there are at least $\frac{d}{2}$ entries that are zero. Consequently, each $w_s$ misclassifies at least $\frac{d}{2}$ standard basis vectors.

## 6 FURTHER RELATED WORK

**Stochastic Adaptive Methods.**   In the convex setting, Duchi et al. (2011) gave a convergence proof for Adagrad, albeit under the strong assumption that the optimization problem was constrained within a compact convex set. Under a similar boundedness assumption, Reddi et al. (2019) showed a variant of Adam, AMSGrad, obtained sublinear regret for online stochastic convex optimization.

In the nonconvex setting, Li & Orabona (2019) was the first to show convergence of AdaGrad. Later, Ward et al. (2021) obtained $\mathcal{O}(\frac{\ln(t)}{t})$ convergence rates for a variant of AdaGrad, AdaGrad-Norm, which uses a uniform scalar for normalizing the gradients. Shen et al. (2023) extended the convergence guarantees to the vector case. Défossez et al. (2022); Zou et al. (2019) obtained $\mathcal{O}(\frac{\ln(t)}{t})$ convergence rates for Adam. These rates are not easily related to the batch rates here, which on the one hand are under a more stringent criterion (margin maximization, which translates to exponentially fast rates for the loss), but on the other hand invoke the additional assumption of linear separability.

**Margin Maximization of gradient methods.**   Margin maximization for linear models was first shown for coordinate descent Zhang & Yu (2005); Telgarsky (2013), and only much later for gradient descent (Soudry et al., 2018; Ji & Telgarsky, 2019a). The idea to use the dual view on purely primal methods such as gradient descent is not new (Molinari et al., 2021; Apidopoulos et al., 2023), and has been applied to simplify and strengthen the aforementioned margin maximization proofs (Ji & Telgarsky, 2019b). Prior work does not seem to have applied these tools to normalized methods such as those considered here.

## 7 CONCLUDING REMARKS AND OPEN PROBLEMS

Connecting to the dual places a different perspective on gradient normalization; rather than drawing roots in AROW and AdaGrad to minimize an online bound, it comes from transformations to a batch dual problem. This perspective allows many further ways to study and improve Adam.

Firstly, the rates here are only $\mathcal{O}(1/t)$ for Normalized GD. Is there a way to recover this bound for ($\ell_\infty$) SignGD? An even more interesting option is that SignGD inherently exhibits nonsmoothness and its rate is stuck at $\mathcal{O}(1/\sqrt{t})$, but under certain favorable conditions, the rate of *Adam*, owing to beneficial smoothing effects of the exponentially-weighted averaging, is in fact $\mathcal{O}(\frac{1}{t})$.

A second natural candidate is to improve the stochastic bounds for Adam; here, the behavior of Adam is only provided a sanity check, and the exponentially-weighted averages do not provide benefits. Is there some way, for specific functions, to benefit from the smoothing effects of the exponentially-weighted averaging scheme?

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

## A    ORGANIZATION

The appendix is organized as follows.

1. In Appendix B, we establish various technical lemmas that will be used in the proofs of both Theorems 29 and 34.

2. In Appendix C, we establish regret guarantees for mirror descent which only require local strong convexity of the mirror potential.

3. In Appendix D, we prove margin maximization.

4. In Appendix E, we prove high probability convergence guarantees.

Before proceeding, we define additional notation that will be used throughout the appendix. Fix a coordinate $j \in [d]$, and let the $m_{sj}$ and $v_{sj}$ denote the averaged first and second moments of gradients respectively:

$$m_{sj} := \frac{1-\beta_1}{1-\beta_1^{s+1}} \sum_{k=0}^{s} \beta_1^{s-k} \nabla \widehat{\mathcal{R}}_k(w_k)_j$$

$$v_{sj} := \frac{1-\beta_2}{1-\beta_2^{s+1}} \sum_{k=0}^{s} \beta_2^{s-k} \nabla \widehat{\mathcal{R}}_k(w_k)_j^2$$

## B    TECHNICAL LEMMAS

The following lemma establishes a $\ell_\infty$ control on the Adam update.

**Lemma 7.** *For any choices of $\beta_1$ and $\beta_2$ such that $\beta_1 \leq \beta_2 < 1$, the Adam update $g_s$, meaning eq. (1), satisfies*

$$\|g_s\|_\infty \leq C := \sqrt{\frac{1-\beta_1}{1-\beta_2}}. \tag{8}$$

*Proof.* For any coordinate $j$, by Jensen's inequality, and since $\beta_1 \leq \beta_2$,

$$\left| m_{sj} \right| = \left| \sum_{k \leq s} a_{s,k} \nabla \widehat{\mathcal{R}}_k(w_k)_j \right| \leq \sum_{k \leq s} a_{s,k} \left| \nabla \widehat{\mathcal{R}}_k(w_k)_j \right|$$

$$\leq \sqrt{\sum_{k \leq s} a_{s,k} \left( \nabla \widehat{\mathcal{R}}_k(w_k)_j \right)^2}$$

$$= \sqrt{\frac{1-\beta_1}{1-\beta_1^{s+1}} \sum_{k \leq s} \beta_1^{s-k} \left( \nabla \widehat{\mathcal{R}}_k(w_k)_j \right)^2}$$

$$\leq \sqrt{\frac{1-\beta_1}{1-\beta_1^{s+1}} \sum_{k \leq s} \beta_2^{s-k} \left( \nabla \widehat{\mathcal{R}}_k(w_k)_j \right)^2}$$

$$= \sqrt{\frac{1-\beta_1}{1-\beta_1^{s+1}} \frac{1-\beta_2^{s+1}}{1-\beta_2} \sum_{k \leq s} b_{s,k} \left( \nabla \widehat{\mathcal{R}}_k(w_k)_j \right)^2}$$

$$\leq \sqrt{\frac{1-\beta_1}{1-\beta_2}} \sqrt{\sum_{k \leq s} b_{s,k} \left( \nabla \widehat{\mathcal{R}}_k(w_k)_j \right)^2}$$

$$= \sqrt{\frac{1-\beta_1}{1-\beta_2}} \sqrt{v_{sj}}.$$

Therefore, for $C := \sqrt{\frac{1-\beta_1}{1-\beta_2}}$, applying the preceding inequality,

$$\left|g_{sj}\right| = \frac{\left|m_{sj}\right|}{\sqrt{v_{sj}} + \epsilon} \leq \frac{\left|m_{sj}\right|}{\sqrt{v_{sj}}} \leq \frac{C\sqrt{v_{sj}}}{\sqrt{v_{sj}}} = C.$$

Therefore, $\left\|g_s\right\|_\infty \leq C$. □

**Remark 8.** *It is interesting to consider an $\ell_2$ variant of Adam, $\ell_2$-Adam, where the normalization in the update consists of $\ell_2$ norms of the gradients. Concretely, suppose the weights $w_s$ are updated using the recurrence, $w_{s+1} = w_s - \eta g_s$, where the update $g_s$ is defined as*

$$g_s := \frac{\sum_{k \leq s} a_{s,k} \nabla \widehat{\mathcal{R}}_k(w_k)}{\sqrt{\sum_{k \leq s} b_{s,k} \left\|\nabla \widehat{\mathcal{R}}_k(w_k)\right\|^2} + \epsilon}.$$

*Indeed, as most of the proofs regarding Adam uses a per-coordinate analysis, many of the same proofs can modified with very little work to handle $\ell_2$-Adam. Like Adam, the type of normalization induces the implicit bias; it can be shown that $\ell_2$-Adam maximizes the $\ell_2$-margin at a rate of $\mathcal{O}(\frac{1}{\sqrt{t}})$. Furthermore, unlike in Adam, the stability constant $\epsilon$ does not change its implicit bias.*

Now we introduce several technical inequalities used in our proofs.

**Lemma 9.** *Suppose $\{\lambda_k\}_{k=0}^s, \{a_k\}_{k=0}^s, \{b_s\}_{k=0}^s$ are nonnegative sequences. Then it follows that*

$$\left|\sqrt{\sum_{k=0}^s \lambda_k a_k^2} - \sqrt{\sum_{k=0}^s \lambda_k b_k^2}\right| \leq \sum_{k=0}^s \sqrt{\lambda_k}\left|a_k - b_k\right|.$$

*Proof.* By expanding the square, Cauchy-Schwarz, and recalling that $\{\lambda_k\}_{k=0}^s, \{a_k\}_{k=0}^s, \{b_s\}_{k=0}^s$ are nonnegative sequences,

$$\left(\sqrt{\sum_{k=0}^s \lambda_k a_k^2} - \sqrt{\sum_{k=0}^s \lambda_k a_k^2}\right)^2 = \sum_{k=0}^s \lambda_k \left(a_k^2 + b_k^2\right) - 2\sqrt{\sum_{k=0}^s \lambda_k a_k^2}\sqrt{\sum_k \lambda_k b_k^2}$$

$$\leq \sum_{k=0}^s \lambda_k \left(a_k^2 + b_k^2\right) - 2\sum_{k=0}^s \lambda_k a_k b_k$$

$$\leq \sum_{k=0}^s \lambda_k \left(a_k - b_k\right)^2.$$

Applying square root on both sides and noting that $\sqrt{\cdot}$ is subadditive grants the desired inequality. □

**Lemma 10.** *For any $x \in \mathbb{R}$,*
$$\left|e^x - 1\right| \leq e^{|x|} - 1.$$

*Proof.* We consider two cases.

1. If $x > 0$,
$$\left|e^x - 1\right| = e^x - 1 = e^{|x|} - 1.$$

2. Suppose $x \leq 0$. For any $x \in \mathbb{R}$, it follows that $(e^x - 1)^2 \geq 0$ which implies $2e^x - e^{2x} \leq 1$. Multiplying $e^{-x}$ on both sides and then subtracting 1 on both sides grants,
$$1 - e^x \leq e^{-x} - 1 = e^{|x|} - 1.$$

To conclude the proof, note that $1 - e^x = \left|e^x - 1\right|$ for $x \leq 0$.

□

**Lemma 11.** *For any $a, b \in \mathbb{R}$,*

$$\left| \frac{\ell'(b)}{\ell'(a)} - 1 \right| \leq e^{|b-a|} - 1.$$

*Proof.* By Theorem 10,

$$\left| \frac{\ell'(b)}{\ell'(a)} - 1 \right| = \left| \frac{e^a + 1}{e^b + 1} - 1 \right| = \left| \frac{e^a - e^b}{e^b + 1} \right| \leq \left| \frac{e^a - e^b}{e^b} \right| = \left| e^{a-b} - 1 \right| \leq e^{|a-b|} - 1.$$

$\square$

**Corollary 12.** *Suppose $w_s$ are updated using eq. (1) and let $C := \frac{1}{\sqrt{1-\beta_2}}$. Then for any $z$ such that $\|z\|_1 \leq R$, and time $k \leq s$,*

$$\left| \frac{\ell'(\langle w_k, z \rangle)}{\ell'(\langle w_s, z \rangle)} - 1 \right| \leq e^{CR \sum_{\tau=k}^{s-1} \eta_\tau} - 1.$$

*Proof.* By Hölder's inequality, triangle inequality, and Lemma 7,

$$\left| \langle w_k - w_s, z \rangle \right| \leq \|w_k - w_s\|_\infty \|z\|_1 = \left\| \sum_{\tau=k}^{s-1} \eta_\tau g_\tau \right\|_\infty \|z\|_1 \leq CR \sum_{\tau=k}^{s-1} \eta_\tau.$$

By the preceding inequality and Theorem 11,

$$\left| \frac{\ell'(\langle w_k, z \rangle)}{\ell'(\langle w_s, z \rangle)} - 1 \right| \leq e^{|\langle w_k, z \rangle - \langle w_s, z \rangle|} - 1 \leq e^{CR \sum_{\tau=k}^{s-1} \eta_\tau} - 1.$$

$\square$

**Lemma 13.** *Suppose $f : [0, \infty) \to [0, \infty)$ is a differentiable function and there exists $\tau \in [0, \infty)$ such that*

$$\forall x \leq \tau, \ f'(x) \geq 0 \quad and \quad \forall x \geq \tau, \ f'(x) \leq 0.$$

*Then*

$$\sum_{k=0}^{\infty} f(k) \leq 2 \int_{k=0}^{\infty} f(k) \, dk.$$

*Proof.* Let $N = \lfloor \tau \rfloor$. Since $f$ is nondecreasing on $[0, N]$, $f$ is nonincreasing on $[N+1, \infty)$, and $f \geq 0$ on the interval $[0, \infty)$,

$$\sum_{k=0}^{\infty} f(k) = \sum_{k=0}^{N} f(k) + \sum_{k=N+1}^{\infty} f(k)$$

$$\leq \sum_{k=0}^{N} \int_{k}^{k+1} f(x) \, dx + \sum_{k=N+1}^{\infty} \int_{k-1}^{k} f(x) \, dx$$

$$= \int_{0}^{N+1} f(x) \, dx + \int_{N}^{\infty} f(x) \, dx$$

$$= \int_{0}^{\infty} f(x) \, dx + \int_{N}^{N+1} f(x) \, dx$$

$$\leq 2 \int_{0}^{\infty} f(x) \, dx.$$

$\square$

**Corollary 14.** *For any $\beta \in (0, 1)$ and $\ell \geq 1$,*

$$\sum_{k=0}^{\infty} \beta^k k^\ell \leq 2 \int_{0}^{\infty} \beta^k k^\ell \, dk.$$

*Proof.* Let $f$ be the function $f : [0, \infty) \to [0, \infty)$ such that $f(k) := \beta^k k^\ell$. By calculation, $f'(k) = \beta^k k^{\ell-1} \left( \ell - k \cdot \ln(\frac{1}{\beta}) \right)$. Abbreviating $\tau := \frac{\ell}{\ln(\frac{1}{\beta})}$,

$$\begin{cases} f'(k) \geq 0, & k \leq \tau, \\ f'(k) \leq 0, & k > \tau. \end{cases}$$

Applying Theorem 13 gives the desired inequality. $\qquad\square$

**Lemma 15.** *For any $\alpha > 0$, $\beta \in (0, 1)$, and learning rates $\eta_s \leq \eta \leq \frac{1-\beta}{2\alpha}$,*

$$\sum_{k=0}^{s} \beta^{s-k} \left( \exp\left[ \alpha \sum_{\tau=k}^{s-1} \eta_\tau \right] - 1 \right) \leq \frac{4\alpha\eta}{(1-\beta)^2}.$$

*Proof.* By change of indices and since $\eta_k \leq \eta$ for all $k$,

$$\sum_{k=0}^{s} \beta^{s-k} \left( \exp\left[ \alpha \sum_{\tau=k}^{s-1} \eta_\tau \right] - 1 \right) = \sum_{k=0}^{s} \beta^k \left( \exp\left[ \alpha \sum_{\tau=1}^{k} \eta_{s-\tau} \right] - 1 \right)$$

$$\leq \sum_{k=0}^{s} \beta^k \left( \exp[\alpha k \eta] - 1 \right)$$

$$= \sum_{k=0}^{s} \beta^k \left( \sum_{\ell=0}^{\infty} \frac{[\alpha k \eta]^\ell}{\ell!} - 1 \right)$$

$$= \sum_{k=0}^{s} \beta^k \left( \sum_{\ell=1}^{\infty} \frac{[\alpha k \eta]^\ell}{\ell!} \right).$$

Switching the order of the summation, and since $\sum_{k=0}^{\infty} \beta^k k^\ell \leq 2 \int_0^\infty \beta^k k^\ell \, dk$ by Theorem 13,

$$\sum_{k=0}^{s} \beta^{s-k} \left( \exp\left[ \alpha \sum_{\tau=k}^{s-1} \eta_\tau \right] - 1 \right) \leq \sum_{k=0}^{s} \beta^k \left( \sum_{\ell=1}^{\infty} \frac{[\alpha k \eta]^\ell}{\ell!} \right)$$

$$\leq \sum_{\ell=1}^{\infty} \frac{[\alpha\eta]^\ell}{\ell!} \sum_{k=0}^{\infty} \beta^k k^\ell$$

$$\leq 2 \sum_{\ell=1}^{\infty} \frac{[\alpha\eta]^\ell}{\ell!} \int_0^\infty \beta^k k^\ell \, dk.$$

Using the fact that $\int_0^\infty \beta^k k^\ell \, dk = \frac{\ell!}{\ln(1/\beta)^{\ell+1}} \leq \frac{\ell!}{(1-\beta)^{\ell+1}}$ and $\alpha\eta \leq \frac{1-\beta}{2}$,

$$\sum_{k=0}^{s} \beta^{s-k} \left( \exp\left[ \alpha \sum_{\tau=k}^{s-1} \eta_\tau \right] - 1 \right) \leq \frac{2}{1-\beta} \sum_{\ell=1}^{\infty} \left( \frac{[\alpha\eta]}{1-\beta} \right)^\ell$$

$$= \frac{2}{1-\beta} \left( \frac{1}{1 - \frac{\alpha\eta}{1-\beta}} - 1 \right)$$

$$= \frac{2}{1-\beta} \left( \frac{\alpha\eta}{1-\beta-\alpha\eta} \right)$$

$$\leq \frac{4\alpha\eta}{(1-\beta)^2}.$$

$\qquad\square$

The following lemma shows that the averaged moments of the gradients is close to the current moment of the gradients. This lemma is critical in both the proof of margin maximization in the deterministic setting (Theorem 29) and low test error guarantee in the stochastic setting (Theorem 34).

**Lemma 16.** *Fix $\beta_1, \beta_2 \in [0, 1)$ and assume the iterates $w_s$ are updated using eq. (1). Fix distribution $\mathcal{D}$ and assume $z \sim \mathcal{D}$ satisfies $\|z\|_1 \leq R$ almost surely. Then for any $\beta \in (0, 1)$ and learning rates $\eta_s$ such that $\eta_s \leq \eta \leq \frac{1-\beta}{2CR}$ where $C = \sqrt{\frac{1-\beta_1}{1-\beta_2}}$ , the average loss derivatives satisfy*

$$\left| \frac{1-\beta}{1-\beta^{s+1}} \sum_{k \leq s} \beta^{s-k} \mathbb{E}_z \ell' \left( \langle w_k, z \rangle \right) - \mathbb{E} \ell' \left( \langle w_s, z \rangle \right) \right| \leq \frac{4CR}{(1-\beta)^2} \eta \mathbb{E}_z \left[ -\ell' \left( \langle w_s, z \rangle \right) \right], \quad (9)$$

*and the averaged first moments of the gradients satisfy*

$$\left\| \frac{1-\beta}{1-\beta^{s+1}} \sum_{k \leq s} \beta^{s-k} \mathbb{E} \left[ \nabla \ell(w_k) \right] - \mathbb{E} \left[ \nabla \ell(w_k) \right] \right\|_1 \leq \frac{4CR^2}{(1-\beta)^2} \eta \mathbb{E}_z \left[ -\ell' \left( \langle w_s, z \rangle \right) \right]. \quad (10)$$

*If instead $\eta \leq \frac{1-\sqrt{\beta}}{2CR}$, then the averaged second moment of the gradients satisfy*

$$\sum_{j \in [d]} \left| \sqrt{\frac{1-\beta}{1-\beta^{s+1}} \sum_{k=0}^{s} \beta^{s-k} \left( \mathbb{E} \ell'(\langle w_k, z \rangle) z_j \right)^2} - \left| \mathbb{E} \ell'(\langle w_s, z \rangle) z_j \right| \right| \leq \frac{CR^2}{\left( 1 - \sqrt{\beta} \right)^2} \eta \mathbb{E}_z \left[ -\ell' \left( \langle w_s, z \rangle \right) \right].$$

$$(11)$$

*Proof.* Applying Theorem 12 and Lemma 15 with $\alpha = CR$,

$$\left| \frac{1-\beta}{1-\beta^{s+1}} \sum_{k \leq s} \beta^{s-k} \mathbb{E}_z \ell' \left( \langle w_k, z \rangle \right) - \mathbb{E}_z \ell' \left( \langle w_s, z \rangle \right) \right|$$

$$\leq \frac{1-\beta}{1-\beta^{s+1}} \sum_{k \leq s} \beta^{s-k} \left| \mathbb{E}_z \left[ \ell' \left( \langle w_k, z \rangle \right) - \ell' \left( \langle w_s, z \rangle \right) \right] \right|$$

$$\leq \frac{1-\beta}{1-\beta^{s+1}} \sum_{k \leq s} \beta^{s-k} \mathbb{E}_z \left[ \left| \ell' \left( \langle w_s, z \rangle \right) \right| \left| \frac{\ell' \left( \langle w_k, z \rangle \right)}{\ell' \left( \langle w_s, z \rangle \right)} - 1 \right| \right]$$

$$\leq \frac{1-\beta}{1-\beta^{s+1}} \mathbb{E}_z \left| \ell' \left( \langle w_s, z \rangle \right) \right| \sum_{k \leq s} \beta^{s-k} \left( \exp \left[ CR \sum_{\tau=k}^{s-1} \eta_\tau \right] - 1 \right)$$

$$\leq \frac{4CR}{(1-\beta)^2} \eta \mathbb{E}_z \left| \ell' \left( \langle w_s, z \rangle \right) \right|.$$

Similarly, applying Theorem 12 and Lemma 15 with $\alpha = CR$,

$$\left\| \frac{1-\beta}{1-\beta^{s+1}} \sum_{k \leq s} \beta^{s-k} \mathbb{E}_z \left[ \nabla \ell(\langle w_k, z \rangle) \right] - \mathbb{E} \left[ \nabla \ell(\langle w_s, z \rangle) \right] \right\|_1$$

$$\leq \frac{1-\beta}{1-\beta^{s+1}} \sum_{k \leq s} \beta^{s-k} \mathbb{E}_z \left[ \left| \ell'(\langle w_k, z \rangle) - \ell'(\langle w_s, z \rangle) \right| \cdot \|z\|_1 \right]$$

$$\leq R \frac{1-\beta}{1-\beta^{s+1}} \sum_{k \leq s} \beta^{s-k} \mathbb{E}_z \left[ \left| \ell' \left( \langle w_s, z \rangle \right) \right| \left| \frac{\ell' \left( \langle w_k, z \rangle \right)}{\ell' \left( \langle w_s, z \rangle \right)} - 1 \right| \right]$$

$$\leq \frac{4CR^2}{(1-\beta)^2} \eta \mathbb{E}_z \left[ -\ell' \left( \langle w_s, z \rangle \right) \right].$$

We now prove eq. (11). Fix $j \in [d]$ and instantiate $\lambda_k, a_k, b_k$ for $k \leq s$ as

$$\lambda_k = \beta^{s-k}, \quad a_k = \left| \mathbb{E}\left[\ell'\left(\langle w_k, z\rangle\right)\right] z_j \right|, \quad b_k = \left| \mathbb{E}\left[\ell'\left(\langle w_s, z\rangle\right)\right] z_j \right|.$$

Applying Theorem 9 with $\lambda_k, a_k, b_k$ defined above, reverse triangle inequality,

$$\left| \sqrt{\sum_{k=0}^{s} \beta^{s-k} \left| \mathbb{E}\left[\ell'\left(\langle w_k, z\rangle\right)\right] z_j \right|} - \sqrt{\sum_{k=0}^{s} \beta^{s-k} \left| \mathbb{E}\left[\ell'\left(\langle w_s, z\rangle\right)\right] z_j \right|} \right|$$

$$\leq \sum_{k=0}^{s} \beta^{(s-k)/2} \left| \mathbb{E}\left[\ell'\left(\langle w_k, z\rangle\right)\right] z_j - \mathbb{E}\left[-\ell'\left(\langle w_s, z\rangle\right)\right] z_j \right|$$

$$\leq \sum_{k=0}^{s} \beta^{(s-k)/2} \mathbb{E}\left[\left|\ell'\left(\langle w_k, z\rangle\right) - \ell'\left(\langle w_s, z\rangle\right)\right| |z_j|\right].$$

Summing over $j \in [d]$ and pushing summation over $j \in [d]$ inside, and recalling $\|z\|_1 \leq R$ almost surely,

$$\sum_{j \in [d]} \left| \sqrt{\sum_{k=0}^{s} \beta^k \left(\mathbb{E}\ell'(\langle w_k, z\rangle)z_j\right)^2} - \sqrt{\sum_{k=0}^{s} \beta^k \left(\mathbb{E}\ell'(\langle w_s, z\rangle)z_j\right)^2} \right|$$

$$\leq \sum_{j \in [d]} \sum_{k=0}^{s} \beta^{(s-k)/2} \mathbb{E}\left[\left|\ell'\left(\langle w_k, z\rangle\right) - \ell'\left(\langle w_s, z\rangle\right)\right| |z_j|\right]$$

$$= \sum_{k=0}^{s} \beta^{(s-k)/2} \mathbb{E}\left[\left|\ell'\left(\langle w_k, z\rangle\right) - \ell'\left(\langle w_s, z\rangle\right)\right| \|z\|_1\right]$$

$$\leq R \sum_{k=0}^{s} \beta^{(s-k)/2} \mathbb{E}\left|\ell'\left(\langle w_k, z\rangle\right) - \ell'\left(\langle w_s, z\rangle\right)\right|$$

$$= R \sum_{k=0}^{s} \beta^{(s-k)/2} \mathbb{E}\left[\left|\ell'\left(\langle w_s, z\rangle\right)\right| \left|\frac{\ell'\left(\langle w_k, z\rangle\right)}{\ell'\left(\langle w_s, z\rangle\right)} - 1\right|\right].$$

Again applying Theorem 12 and Lemma 15 with $\alpha = CR$ and substituting $\beta$ with $\sqrt{\beta}$,

$$\sum_{j \in [d]} \left| \sqrt{\sum_{k=0}^{s} \beta^k \left(\mathbb{E}\ell'(\langle w_k, z\rangle)z_j\right)^2} - \sqrt{\sum_{k=0}^{s} \beta^k \left(\mathbb{E}\ell'(\langle w_s, z\rangle)z_j\right)^2} \right|$$

$$\leq \frac{4CR^2}{\left(1 - \sqrt{\beta}\right)^2} \eta \mathbb{E}\left[-\ell'\left(\langle w_s, z\rangle\right)\right].$$

Finally multiplying both sides by $\sqrt{\frac{1-\beta}{1-\beta^{s+1}}}$ and noting $\sqrt{\frac{1-\beta}{1-\beta^{s+1}}} \leq 1$ grants the desired inequality. $\qquad \square$

**Lemma 17.** *Let $B, \epsilon > 0$ be fixed constants and denote $f(x) := \sum_{j \in [d]} \frac{x_j^2}{x_j + \epsilon}$. Let $S$ be the intersection of the positive orthant and a $\ell_1$ bounded ball,*

$$S := \{u \mid u_j \geq 0 \text{ and } \|u\|_1 \leq B\}.$$

*Then $x = \left(\frac{B}{d}, \dots, \frac{B}{d}\right)$ is the solution to the following optimization problem,*

$$\min_{x \in S} f(x) \quad s.t. \sum_{j \in [d]} x_j = B.$$

*Proof.* First note that S is a convex set and $f$ is a strongly convex function over $S$. Hence, it follows that the optimal solution must be unique. By symmetry, it follows that each coordinate of the optimal solution must be identical. By the constraint condition and feasibility, the optimal solution must be $x = \left( \frac{B}{d}, \ldots, \frac{B}{d} \right)$. $\qquad\square$

The following lemma establishes that SignGD updates are close to Adam updates.

**Lemma 18.** *For any $\epsilon \geq 0$ and $j \in [d]$, denote $\sigma_{sj} := \frac{\nabla R(w_s)_j}{\left| \nabla \widehat{\mathcal{R}}(w_s)_j \right| + \epsilon}$ and for $\epsilon = 0$, take the convention*

*that $\sigma_{sj} = 0$ when $\nabla R(w_s)_j = 0$. Then, under the same setting as Theorem 3, for all $s < t$,*

$$\left| \left\langle \sigma_s - g_s, \nabla \widehat{\mathcal{R}}(w_s) \right\rangle \right| \leq \frac{12 C^2 R^2}{\left( 1 - \sqrt{\beta_2} \right)^2} \eta \widehat{\mathcal{G}}(w_s).$$

*Proof.* Applying Lemma 16 with $\mathcal{D}$ being the uniform distribution over the training data $(z_1, \ldots, z_n)$ (i.e. the empirical distribution), the following inequalities hold,

$$\left\| m_s - \nabla \widehat{\mathcal{R}}(w_s) \right\|_1 \leq \frac{4 C R^2}{\left( 1 - \beta_1 \right)^2} \eta \widehat{\mathcal{G}}(w_s), \tag{12}$$

$$\sum_{j \in [d]} \left| \sqrt{v_{sj}} - \left| \nabla \widehat{\mathcal{R}}(w_s)_j \right| \right| \leq \frac{4 C R^2}{1 - \sqrt{\beta_2}} \eta \widehat{\mathcal{G}}(w_s). \tag{13}$$

Expanding and applying triangle inequality,

$$\left| \left\langle \sigma_s - g_s, \nabla \widehat{\mathcal{R}}(w_s) \right\rangle \right| \leq \left| \sum_j \left( \frac{\nabla \widehat{\mathcal{R}}(w_s)_j}{\left| \nabla \widehat{\mathcal{R}}(w_s)_j \right| + \epsilon} - \frac{m_{sj}}{\sqrt{v_{sj}} + \epsilon} \right) \nabla \widehat{\mathcal{R}}(w_s)_j \right|$$

$$\leq \sum_j \frac{\left| \nabla \widehat{\mathcal{R}}(w_s)_j \sqrt{v_{sj}} - m_{sj} \left| \nabla \widehat{\mathcal{R}}(w_s)_j \right| + \left( \nabla \widehat{\mathcal{R}}(w_s)_j - m_{sj} \right) \epsilon \right|}{\left( \left| \nabla \widehat{\mathcal{R}}(w_s)_j \right| + \epsilon \right) \left( \sqrt{v_{sj}} + \epsilon \right)} \left| \nabla \widehat{\mathcal{R}}(w_s)_j \right|$$

$$\leq \sum_j \frac{\left| \nabla \widehat{\mathcal{R}}(w_s)_j \sqrt{v_{sj}} - m_{sj} \left| \nabla \widehat{\mathcal{R}}(w_s)_j \right| + \left( \nabla \widehat{\mathcal{R}}(w_s)_j - m_{sj} \right) \epsilon \right|}{\sqrt{v_{sj}} + \epsilon}$$

$$= \sum_j \frac{\left| \left( \nabla \widehat{\mathcal{R}}(w_s)_j - m_{sj} \right) \sqrt{v_{sj}} + m_{sj} \left( \sqrt{v_{sj}} - \left| \nabla \widehat{\mathcal{R}}(w_s)_j \right| \right) + \left( \nabla \widehat{\mathcal{R}}(w_s)_j - m_{sj} \right) \epsilon \right|}{\sqrt{v_{sj}} + \epsilon}$$

$$\leq \sum_j \frac{\left| \nabla \widehat{\mathcal{R}}(w_s)_j - m_{sj} \right| \sqrt{v_{sj}} + m_{sj} \left| \sqrt{v_{sj}} - \left| \nabla \widehat{\mathcal{R}}(w_s)_j \right| \right| + \left| \nabla \widehat{\mathcal{R}}(w_s)_j - m_{sj} \right| \epsilon}{\sqrt{v_{sj}} + \epsilon}$$

$$\leq \sum_j \left| \nabla \widehat{\mathcal{R}}(w_s)_j - m_{sj} \right| + C \left| \sqrt{v_{sj}} - \left| \nabla \widehat{\mathcal{R}}(w_s)_j \right| \right| + \left| \nabla \widehat{\mathcal{R}}(w_s)_j - m_{sj} \right|$$

$$= \sum_j 2 \left| \nabla \widehat{\mathcal{R}}(w_s)_j - m_{sj} \right| + C \left| \sqrt{v_{sj}} - \left| \nabla \widehat{\mathcal{R}}(w_s)_j \right| \right|$$

$$= 2 \left\| \nabla \widehat{\mathcal{R}}(w_s) - m_s \right\|_1 + C \sum_j \left| \sqrt{v_{sj}} - \left| \nabla \widehat{\mathcal{R}}(w_s)_j \right| \right|.$$

By eq. (12) and eq. (13), since $\beta_1 \leq \beta_2 \leq \sqrt{\beta_2}$, and $C \geq 1$,

$$\left| \left\langle \sigma_s - g_s, \nabla \widehat{\mathcal{R}}(w_s) \right\rangle \right| \leq \frac{12 C^2 R^2}{\left( 1 - \sqrt{\beta_2} \right)^2} \eta \widehat{\mathcal{G}}(w_s).$$

$\square$

The following lemma is the well known Titu's lemma. For the sake of completeness, we provide the proof.

**Lemma 19.** *Let $\{a_j\}_{j\in d}, \{b_j\}_{j\in d}$ be nonnegative sequences of real numbers. Then*

$$\sum_{j\in[d]} \frac{a_j^2}{b_j} \geq \frac{\left(\sum_{j\in[d]} a_j\right)^2}{\sum_{j\in[d]} b_j}.$$

*Proof.* By Cauchy-Schwarz,

$$\left(\sum_{j\in[d]} a_j\right)^2 = \left(\sum_{j\in[d]} \frac{a_j}{\sqrt{b_j}} \cdot \sqrt{b_j}\right)^2 \leq \left(\sum_{j\in[d]} b_j\right) \cdot \sum_{j\in[d]} \frac{a_j^2}{b_j}.$$

Dividing both sides by $\sum_{j\in[d]} b_j$ gives the desired claim. $\square$

The last lemma of this section gives an equivalency between the primal and dual objectives, providing the crucial insight on how to obtain margin maximization guarantees.

**Lemma 20.** *For any norm $\|\cdot\|$ and any matrix $Z$ encoding a linearly separable problem (there exists $v$ with $Zv > 0$), then*

$$\min_{-\psi^*(q)\leq 0} \frac{1}{2}\|Z^\mathsf{T}q\|^2 = -\inf_{w\in\mathbb{R}^d, r>0} r\psi(Z\frac{w}{r}) + \frac{1}{2}\|w\|_*^2.$$

*Proof.* By computation, the convex conjugate of the function $f(u) = \frac{1}{2}\|u\|^2$ is $f^*(v) = \frac{1}{2}\|v\|_*^2$. The rest of the proof proceeds as the duality in Ji & Telgarsky (2019b, Proof of Theorem A.2), except using $f$ and $f^*$ in place of $\|\cdot\|_2$ and $\iota_{\|\cdot\|(x)}$ which is 0 when $\|x\|_2 \leq 1$ and $\infty$ otherwise. $\square$

## C  MIRROR DESCENT ANALYSIS

The following lemma establishes a mirror descent regret guarantee using only local strong convexity of the mirror potential.

**Lemma 21.** *Given initial iterates $u_0, v_0 \in \mathbb{R}^d$, learning schedule $\{\eta_s\}_{s<t} \subset \mathbb{R}$, and a sequence $\{\xi_s\} \subset \mathbb{R}^d$, suppose the iterates $v_s$ follow the update rule,*

$$v_{s+1} = v_s - \eta_s\xi_s.$$

*Let $h$ be any closed, proper, convex function and let $h^*$ be the convex conjugate of $h$. Suppose $u_s = \nabla h^*(v_s)$. Then, the sequence $\{u_s\}_{s<t}$ can equivalently be generated by mirror descent,*

$$u_{s+1} = \arg\min\{\eta_s \langle \xi_s, u\rangle + D_h(u, u_s) : u \in \mathbb{R}^d\}.$$

*In addition, if $h$ is $\lambda_s^{-1}$ strongly convex over the line segment $[u_s, u_{s+1}]$ with respect to $\|\cdot\|$,*

$$h^*(v_0) - h^*(v_t) \geq \sum_{s<t}\eta_s \langle \xi_s, u_s\rangle - \frac{\eta_s^2\lambda_s}{2}\|\xi_s\|_*^2.$$

*Proof.* By convexity, since $u_s = \nabla h^*(v_s)$, we have $v_s = \nabla h(u_s)$ and hence the primal update $v_{s+1} = v_s - \eta_s\xi_s$ gives the following induced dual update

$$\nabla h(u_{s+1}) = \nabla h(u_s) - \eta\xi_s,$$

which is equivalent to the mirror descent update described in Lemma 21.

Then, by the equality case of Fenchel-Young, local strong convexity of $h$, and another application of Fenchel-Young inequality,

$$
\begin{aligned}
h^*(v_0) - h^*(v_t) &= \sum_{s=0}^{t-1} h^*(v_s) - h^*(v_{s+1}) \\
&= \sum_{s=0}^{t-1} h(u_{s+1}) - \langle v_{s+1}, u_{s+1} \rangle + h^*(v_s) \\
&= \sum_{s=0}^{t-1} D_h(u_{s+1}, u_s) + \langle v_s - v_{s+1}, u_{s+1} \rangle \\
&= \sum_{s=0}^{t-1} D_h(u_{s+1}, u_s) + \langle \eta_s \xi_s, u_{s+1} \rangle \\
&\geq \sum_{s=0}^{t-1} \frac{1}{\lambda_s} \|u_{s+1} - u_s\| + \langle \eta_s \xi_s, u_{s+1} - u_s \rangle + \langle \eta_s g_s, u_s \rangle \\
&\geq \sum_{s=0}^{t-1} \frac{1}{2\lambda_s} \|u_{s+1} - u_s\| - \frac{\lambda_s}{2} \|\eta_s \xi_s\|_*^2 - \frac{1}{2\lambda_s} \|u_{s+1} - u_s\| + \langle \eta_s \xi_s, u_s \rangle \\
&= \sum_{s=0}^{t-1} \langle \eta_s \xi_s, u_s \rangle - \frac{\lambda_s}{2} \|\eta_s \xi_s\|_*^2.
\end{aligned}
$$

$\square$

From the mirror descent result, we can develop a regret guarantee with $h = \overline{\mathcal{R}}^*$. We first show that the local strong convexity condition is satisfied.

**Lemma 22.** *Fix $\beta_1, \beta_2 \in (0,1)$ and assume the iterates $w_s$ are updated using eq. (1). Suppose $\eta_s \leq \frac{1}{2C}$ where $C = \sqrt{\frac{1-\beta_1}{1-\beta_2}}$. Abbreviating $\lambda_s := 2\overline{G}(w_s)$, the following statements hold.*

*1. On the line segment $[Zw_s, Zw_{s+1}]$, the risk $\overline{\mathcal{R}}$ is $\lambda_s$ smooth with respect to $\ell_\infty$ norm.*

*2. On the line segment $\left[\nabla \overline{\mathcal{R}}(Zw_s), \nabla \overline{\mathcal{R}}(Zw_{s+1})\right] = [\bar{q}_s, \bar{q}_{s+1}]$, the conjugate of the loss $\overline{\mathcal{R}}^*$ is $\lambda_s^{-1}$ strongly convex with respect to the $\ell_1$ norm.*

*Proof.* For any $w$, since $|\ell''| \leq |\ell'|$,

$$
\left\| \nabla \overline{G}(w) \right\|_1 = \left\| \frac{1}{N} \sum_{i \in [N]} -\ell''(\langle w, \bar{z}_i \rangle) \bar{z}_i \right\|_1 \leq \overline{G}(w) \max_i \|\bar{z}_i\|_1 \leq \overline{G}(w). \tag{14}
$$

Now let $w_c := (1-c)w_s + cw_{s+1}$ and let $\alpha \in [0,1]$ be the constant such that

$$
\overline{G}(w_\alpha) = \sup_{c \in [0,1]} \overline{G}(w_c).
$$

By the mean value theorem, there exists $r \in (0, \alpha)$ such that $\overline{G}(w_\alpha) - \overline{G}(w_s) = \langle \nabla \overline{G}(w_r), w_\alpha - w_s \rangle$. By eq. (14) and definition of $\alpha$, and noting that $\|w_\alpha - w_s\|_\infty = \alpha \|w_{s+1} - w_s\|_\infty \leq C\eta_s$,

$$
\begin{aligned}
\overline{G}(w_\alpha) &= \overline{G}(w_s) + \overline{G}(w_\alpha) - \overline{G}(w_s) \\
&= \overline{G}(w_s) + \langle \nabla G(w_r), w_\alpha - w_s \rangle \\
&\leq \overline{G}(w_s) + \left\| \nabla \overline{G}(w_r) \right\|_1 \|w_\alpha - w_s\|_\infty \\
&\leq \overline{G}(w_s) + \overline{G}(w_r) \|w_\alpha - w_s\|_\infty \\
&\leq \overline{G}(w_s) + C\eta_s \overline{G}(w_\alpha).
\end{aligned}
$$

Rearranging terms and since $\eta_s C \leq \frac{1}{2}$ by assumption,

$$\overline{G}(w_\alpha) = \frac{1}{1 - C\eta_s}\overline{G}(w_s) \leq 2\overline{G}(w_s). \tag{15}$$

Hence, eq. (15) and eq. (14) implies $\overline{G}$ is $2\overline{G}(w_s)$ lipschitz with respect to $\ell_\infty$ on the interval $[w_s, w_{s+1}]$. Therefore, since $\overline{G}$ is lipschitz, applying eq. (15), for any $w, w'$ on the line segment $[w_s, w_{s+1}]$,

$$\left\|\nabla\overline{\mathcal{R}}(Zw) - \nabla\overline{\mathcal{R}}(Zw')\right\|_1 = \left\|\frac{1}{N}\sum_i \left(\ell'(\langle w, \overline{z}_i\rangle) - \ell'(\langle w', \overline{z}_i\rangle)\right)\overline{z}_i\right\|_1$$

$$\leq \left|\overline{G}(w) - \overline{G}(w')\right|\max_i\|\overline{z}_i\|_1$$

$$\leq \left|\overline{G}(w) - \overline{G}(w')\right|$$

$$\leq 2\overline{G}(w_s)\left\|w - w'\right\|_\infty.$$

Hence, $\overline{\mathcal{R}}$ is $\lambda_s$ smooth with respect to $\ell_\infty$ norm which implies $\overline{\mathcal{R}}^*$ is $\lambda_s^{-1}$ is strongly convex with respect to the $\ell_1$ norm. $\qquad\square$

**Lemma 23.** $-\psi$ is $\frac{2}{\gamma}\|Z^\mathsf{T}q_s\|_1$ smooth with respect to $\|\cdot\|_\infty$ along the line segment $[w_s, w_{s+1}]$.

*Proof.* Similar to the proof of Ji & Telgarsky (2019b, Lemma 5.3), to check that $-\psi$ is $\beta$ smooth with respect to $\ell_\infty$ norm on a convex set $S$, we only to ensure for all $\xi \in S$ and $v \in \mathbb{R}^n$,

$$\sum_{i=1}^n \frac{\ell''(\xi_i)}{-\ell'(\psi(\xi))}v_i^2 \leq \beta\max_i v_i^2.$$

Take any $\xi \in [Zw_s, Zw_{s+1}]$ and let $w \in [w_s, w_{s+1}]$ such that $\xi = Zw$. Then,

$$\sum_{i=1}^n \frac{\ell''(\xi_i)}{-\ell'(\psi(\xi))}v_i^2 \leq \max_i v_i^2 \sum_{i=1}^n \frac{\ell''(\xi_i)}{-\ell'(\psi(\xi))}$$

$$\leq \sum_{i=1}^n \frac{-\ell'(\xi_i)}{-\ell'(\psi(\xi))}$$

$$= n\frac{G(w)}{-\ell'(\psi(\xi))}$$

$$\leq n\frac{2G(w_s)}{-\ell'(\psi(\xi))}$$

$$\leq n\frac{2G(w_s)}{-\ell'(\psi(Zw_s))}$$

$$\leq \frac{2\|Z^\mathsf{T}q_s\|_1}{\gamma}.$$

$\qquad\square$

The following lemma establishes a regret guarantee for mirror descent where the mirror potential is $h := \overline{\mathcal{R}}^*$.

**Lemma 24.** *Suppose the iterates $w_s$ are generated using the update rule eq. (1) where $\|g_s\|_\infty \leq C$ and $\eta_s \leq \eta \leq \frac{1}{2C}$. Then,*

$$\overline{\mathcal{R}}(w_0) - \overline{\mathcal{R}}(w_t) \geq \sum_{s<t}\eta_s\left(\left\langle\overline{Z}g_s, \overline{q}_s\right\rangle - \eta_s\overline{G}(w_s)\left\|\overline{Z}g_s\right\|_\infty^2\right).$$

*Proof.* By Lemma 22, the function $h := \overline{\mathcal{R}}^*$ is $\lambda_s^{-1}$ strongly convex on the interval $[\bar{q}_s, \bar{q}_{s+1}]$ which implies

$$D_h(q_{s+1}, \bar{q}_s) \geq \frac{1}{2\lambda_s} \|q_{s+1} - \bar{q}_s\|_1^2.$$

By the preceding inequality and applying Fenchel-Young,

$$-D_h(\bar{q}_{s+1}, \bar{q}_s) + \left\langle \eta_s \overline{Z} g_s, \bar{q}_s - \bar{q}_{s+1} \right\rangle \leq -\frac{1}{2\lambda_s} \|q_{s+1} - \bar{q}_s\|_1^2 + \frac{\lambda_s}{2} \left\| \eta_s \overline{Z} g_s \right\|_\infty^2 + \frac{\|\bar{q}_s - q_{s+1}\|_1^2}{2\lambda_s}$$

$$\leq \frac{\lambda_s}{2} \left\| \eta_s \overline{Z} g_s \right\|_\infty^2.$$

Now by the equality case of Fenchel - Young and since $\bar{q}_s = \nabla h^*(\bar{p}_s)$ and by the preceding inequality,

$$h^*(p_0) - h^*(p_t) = \sum_{s<t} h^*(\bar{p}_s) - h^*(p_{s+1})$$

$$= \sum_{s<t} \langle \bar{p}_s, \bar{q}_s \rangle - h(\bar{q}_s) - h^*(p_{s+1})$$

$$= \sum_{s<t} D_h(\bar{q}_{s+1}, \bar{q}_s) + \left\langle \eta_s \overline{Z} g_s, \bar{q}_{s+1} \right\rangle$$

$$= \sum_{s<t} D_h(\bar{q}_{s+1}, \bar{q}_s) + \left\langle \eta_s \overline{Z} g_s, \bar{q}_{s+1} - \bar{q}_s \right\rangle + \left\langle \eta_s \overline{Z} g_s, \bar{q}_s \right\rangle$$

$$\geq \sum_{s<t} \left\langle \eta_s \overline{Z} g_s, \bar{q}_s \right\rangle - \frac{\lambda_s}{2} \left\| \eta_s \overline{Z} g_s \right\|_\infty^2.$$

Recalling that $\lambda_s := 2\overline{G}(w_s)$ completes the proof. $\qquad\square$

## D  MARGIN MAXIMIZATION

In this section, we prove several margin maximization results for different full-batch methods. The following lemma gives a proof of the SignGD claim in Theorem 2.

**Lemma 25.** *Under the same assumptions as Theorem 2, for $t \geq \frac{8n^2}{\gamma_\infty^2}$, SignGD maximizes the $\ell_\infty$-margin:*

$$\frac{\psi(Zw_t)}{\|w_t\|_\infty} \geq \gamma_\infty - \frac{8n}{\gamma_\infty \sqrt{t}}.$$

*Proof.* We first invoke Lemma 1 with the following instantiation: mirror potential $h = (-\psi)^*$, primal iterates $v_s = Zw_s$, dual iterates $u_s = q_s$, primal update $g_s = \text{sign}(\nabla \widehat{\mathcal{R}}(w_s))$, dual update $\xi_s = Zg_s$, and learning rate $\eta = \frac{\gamma_\infty}{4\sqrt{t}}$.

Hence, by Lemma 1 and since by Lemma 20 the dual objective satisfies $\|Z^\mathsf{T} q_s\|_1 \geq \gamma_\infty$,

$$\psi(Zw_t) \geq \sum_{s<t} \eta \left[ \langle Zg_s, q_s \rangle - \frac{\eta \|Z^\mathsf{T} q_s\|_1}{2\gamma} \|Zg_s\|_\infty^2 \right] + \psi(Zw_0)$$

$$\geq \sum_{s<t} \eta \left( 1 - \frac{1}{8\sqrt{t}} \right) \|Z^\mathsf{T} q_s\|_1 + \psi(Zw_0)$$

$$\geq \sum_{s<t} \eta \left( 1 - \frac{1}{8\sqrt{t}} \right) \gamma_\infty + \psi(Zw_0). \tag{16}$$

As $w_0 = 0$, the inital smoothed margin satisfies $\psi(Zw_0) = -\ell^{-1}(n \ln(2)) \geq -n$. By the choice of the learning rate and since $t \geq \frac{8n^2}{\gamma_\infty^2}$, the right hand side of eq. (16) is nonnegative.

To finish the proof from here, it suffices to divide $\|w_t\|_\infty$ across both sides of eq. (16), and upper bound $\|w_t\|_\infty$. By a simple application of triangle inequality and noting that each entry of the SignGD

update is bounded by 1, we obtain that $\|w_t\|_\infty \le \sum_{s<t} \eta = \eta t$. Putting it altogether,

$$\frac{\psi(Zw_t)}{\|w_t\|_\infty} \ge \frac{1}{\|w_t\|_\infty} \left( \sum_{s<t} \eta \left( 1 - \frac{1}{8\sqrt{t}} \right) \gamma_\infty - n \right)$$

$$\ge \gamma_\infty - \frac{8n}{\gamma_\infty \sqrt{t}}.$$

□

The following lemma provides a warm start for $\ell_2$-SignGD.

**Lemma 26.** *Under the same assumptions as Theorem 2, for $t \ge t_0 := \frac{4\ln^2(8n)}{\gamma^2}$, the empirical risk satisfies $\widehat{\mathcal{R}}(w_t) < \frac{\ell(0)}{n}$ which implies perfect classification.*

*Proof.* Let $\tau$ be the first time (potentially infinite) such that $\widehat{\mathcal{R}}(w_\tau) < \frac{\ell(0)}{n}$. Let $u$ be the unit vector that has margin $\gamma_2$ in Assumption 1. Let $\bar{u} := \frac{\ln(8n)}{\gamma} u$. Since the exponential loss upper bounds logistic loss and by the definition of $\bar{u}$,

$$\widehat{\mathcal{R}}(\bar{u}) \le \frac{1}{n} \sum_{i\in[n]} \exp(-\langle \bar{u}, z_i \rangle) \le \frac{1}{n} \sum_{i\in[n]} \frac{1}{8n} = \frac{1}{8n} < \frac{\ell(0)}{4n}.$$

Hence, for all time $s \le \tau$, the empirical risk is bounded below:

$$\widehat{\mathcal{R}}(\bar{u}) < \frac{\ell(0)}{4n} \le \frac{\widehat{\mathcal{R}}(w_s)}{4}. \tag{17}$$

In addition, for all time $s \le \tau$, by the choice of $\eta = \frac{1}{4}$,

$$\frac{\eta}{2} \left\| \nabla \widehat{\mathcal{R}}(w_s) \right\| \le \frac{\widehat{\mathcal{R}}(w_s)}{8}. \tag{18}$$

Expanding the square and by convexity and eqs. (17) and (18), for all time $s \le \tau$,

$$\|w_{s+1} - \bar{u}\|^2 - \|w_s - \bar{u}\|^2 = 2\eta \langle g_s, \bar{u} - w_s \rangle + \eta^2 \|g_s\|^2$$

$$= 2\eta \left\langle \frac{\nabla \widehat{\mathcal{R}}(w_s)}{\left\| \nabla \widehat{\mathcal{R}}(w_s) \right\|}, \bar{u} - w_s \right\rangle + \eta^2 \|g_s\|^2$$

$$\le 2\eta \frac{\widehat{\mathcal{R}}(\bar{u}) - \widehat{\mathcal{R}}(w_s)}{\left\| \nabla \widehat{\mathcal{R}}(w_s) \right\|} + \eta^2$$

$$\le 2\eta \frac{\widehat{\mathcal{R}}(\bar{u}) - \widehat{\mathcal{R}}(w_s) + \frac{\eta}{2} \left\| \nabla \widehat{\mathcal{R}}(w_s) \right\|}{\left\| \nabla \widehat{\mathcal{R}}(w_s) \right\|}$$

$$\le -\eta \frac{\widehat{\mathcal{R}}(w_s)}{\left\| \nabla \widehat{\mathcal{R}}(w_s) \right\|}.$$

Therefore, rearranging terms, summing across time $s \le \tau$, and telescoping terms,

$$\sum_{s<\tau} \frac{\widehat{\mathcal{R}}(w_s)}{\left\| \nabla \widehat{\mathcal{R}}(w_s) \right\|} \le \frac{\|w_0 - \bar{u}\|^2 - \|w_t - \bar{u}\|^2}{\eta} \le \frac{\|\bar{u}\|^2}{\eta} = \frac{4\ln^2(8n)}{\gamma^2}. \tag{19}$$

Note that $\frac{\widehat{\mathcal{R}}(w_s)}{\left\| \nabla \widehat{\mathcal{R}}(w_s) \right\|} \ge 1$. Hence, eq. (19) simplifies,

$$\tau \le \frac{4\ln^2(8n)}{\gamma^2}.$$

To conclude the proof, it suffices to note that the loss is monotonically decreasing.

□

With the warm start result above, we can prove the $\ell_2$-SignGD claim in Theorem 2.

**Lemma 27.** *Under the same setting as Theorem 2, for $t \geq \frac{\ln^2(8n)}{\gamma^2}$, $\ell_2$-SignGD maximizies the $\ell_2$-margin:*

$$\frac{\psi(Zw_t)}{\|w_t\|_2} \geq \gamma_2 - \frac{32\ln(8n)}{\gamma_2^2 t}.$$

*Proof.* By Lemma 26, for all time $t \geq t_0 := \frac{4\ln^2(8n)}{\gamma^2}$, the classifier $w_t$ has perfect classification. Hence, $\psi$ is 2-smooth by Ji & Telgarsky (2019b, Lemma 5.2).

Now we invoke Lemma 1 with the following instantiation: mirror potential $h = (-\psi)^*$, primal iterates $v_s = Zw_s$, dual iterates $u_s = q_s$, primal update $g_s = -\nabla\psi(Zw_s)$, dual update $\xi_s = Zg_s$, and learning rate $\theta_s = \frac{1}{4\|\nabla\psi(Zw_s)\|}$.

Hence, since $\psi(Zw_{t_0}) > 0$,

$$\psi(Zw_t) \geq \theta_t\left(\frac{1}{2}\|Z^\intercal q_t\|^2 - \frac{1}{2}\|Z^\intercal q_0\|^2\right) + \sum_{t_0 \leq s < t} \theta_s \langle Zg_s, q_s\rangle$$

$$\geq \sum_{t_0 \leq s < t} \theta_s\|Z^\intercal q_s\|^2. \tag{20}$$

To finish the proof, we need an upper bound on $\|w_t\|$. First note that Lemma 26 also gives an upper bound on $\|w_{t_0}\|$, namely $\|w_{t_0}\| \leq \|\bar{u}\| + \|w_{t_0} - \bar{u}\| \leq \frac{8\ln(8n)}{\gamma}$. Hence, by triangle inequality,

$$\|w_t\| \leq \|w_{t_0}\| + \|w_{t_0} - w_t\| = \|w_{t_0}\| + \sum_{s=t_0}^{t} \theta_s\|Z^\intercal q_s\|.$$

Therefore,

$$\frac{\psi(Zw_t)}{\|w_t\|} \geq \gamma \frac{\sum_{s=t_0}^{t} \theta_s\|Z^\intercal q_s\|}{\sum_{s=t_0}^{t} \theta_s\|Z^\intercal q_s\| + \|w_{t_0}\|}$$

$$= \gamma\left(1 - \frac{\|w_{t_0}\|}{\|w_{t_0}\| + \sum_{s=t_0}^{t} \theta_s\|Z^\intercal q_s\|}\right)$$

$$\geq \gamma\left(1 - \frac{32\ln(8n)}{\gamma^2 t}\right).$$

$\square$

**Theorem 28.** *Suppose Assumption 1 holds. Take $\|\cdot\|$ to be $\|\cdot\|_\infty$ and $\|\cdot\|_2$ for SignGD and $\ell_2$-SignGD respectively. Assume that for all $i \in [n]$, the data is bounded, $\|x_i\|_* \leq 1$. Suppose the iterates $w_s$ are updated via SignGD, meaning eq. (3), with learning rate $\eta = \frac{\gamma_\infty}{4\sqrt{t}}$. Then for $t \geq \frac{8n^2}{\gamma_\infty^2}$, SignGD maximizes the $\ell_\infty$-margin:*

$$\frac{\psi(Zw_t)}{\|w_t\|_\infty} \geq \gamma_\infty - \frac{8n}{\gamma_\infty \sqrt{t}}.$$

*If the iterates $w_s$ are updated using $\ell_2$-SignGD, meaning eq. (4), with learning rate $\eta = \frac{\gamma_2}{4}$, then for $t \geq \frac{\ln^2(8n)}{\gamma^2}$, $\ell_2$-SignGD maximizies the $\ell_2$-margin:*

$$\frac{\psi(Zw_t)}{\|w_t\|_2} \geq \gamma_2 - \frac{32\ln(8n)}{\gamma_2 t}.$$

*Proof.* Simply apply Lemma 25 and 27. $\square$

The following theorem is the margin maximization result for full-batch Adam, which corresponds to Theorem 3.

**Theorem 29.** *Suppose Assumption 1 holds with $\gamma = \gamma_\infty$ and $\|x_i\|_1 \leq 1$ for every $i \in [n]$. Let $C := \sqrt{\frac{1-\beta_1}{1-\beta_2}}$. For any $0 \leq \beta_1 \leq \beta_2 < 1$ and $0 < \epsilon \leq 1$, for constant learning rate $\eta = \frac{1}{\sqrt{t}}$ where Adam iterates are updated using eq. (1), for all time $t \geq \frac{4096 C^4 n^2}{\gamma^2 (1-\sqrt{\beta_2})^2}$ and $\inf_{s<t} \left\|\nabla\widehat{\mathcal{R}}(w_s)\right\|_1 \geq d\sqrt{\epsilon}$, Adam maximizes the $\ell_\infty$ margin,*

$$\frac{\psi(Zw_t)}{\|w_t\|_\infty} \geq \gamma\left(1 - \sqrt{\epsilon}\right)\sqrt{\frac{1-\beta_2}{1-\beta_1}} - \frac{24C^2 n}{(1-\sqrt{\beta_2})^2 \sqrt{t}}. \tag{21}$$

*Proof.* Applying Lemma 21 with $h = (-\psi)^*$, $v_s = Zw_s$, $\xi_s = Zd_s$, and $u_s = q_s := -\nabla\psi(Zw_s)$, we have the following bound,

$$\psi(Zw_t) - \psi(Zw_0) \geq \sum_{s=0}^{t} \eta \langle Zd_s, q_s \rangle - \frac{\eta^2 n}{2}\|Zd_s\|_\infty^2. \tag{22}$$

We first prove a few preliminary inequalities to handle the inner product term $\langle Zd_s, q_s \rangle$. Fix $s < t$. Recall that $\left\|\nabla\widehat{\mathcal{R}}(w_s)\right\| \geq d\sqrt{\epsilon}$. In addition, note that for any $a > 0$, the function $g(x) = \frac{x}{x+a}$ is monotonically increasing on the positive reals. Hence,

$$\frac{\left\|\nabla\widehat{\mathcal{R}}(w_s)\right\|_1}{\left\|\nabla\widehat{\mathcal{R}}(w_s)\right\|_1 + \epsilon d} \geq \frac{d\sqrt{\epsilon}}{d\sqrt{\epsilon} + d\epsilon} = \frac{\sqrt{\epsilon}}{\sqrt{\epsilon} + \epsilon} = 1 - \frac{\epsilon}{\epsilon + \sqrt{\epsilon}} \geq 1 - \sqrt{\epsilon}.$$

In addition, by Titu's lemma and the preceding inequality,

$$\sum_{j \in [d]} \frac{(\nabla\widehat{\mathcal{R}}(w_s))_j^2}{\left|(\nabla\widehat{\mathcal{R}}(w_s))_j\right| + \epsilon} \geq \frac{\left\|\nabla\widehat{\mathcal{R}}(w_s)\right\|_1^2}{\left\|\nabla\widehat{\mathcal{R}}(w_s)\right\|_1 + d\epsilon}$$

$$= \left\|\nabla\widehat{\mathcal{R}}(w_s)\right\|_1 \frac{\left\|\nabla\widehat{\mathcal{R}}(w_s)\right\|_1}{\left\|\nabla\widehat{\mathcal{R}}(w_s)\right\|_1 + d\epsilon}$$

$$\geq \left\|\nabla\widehat{\mathcal{R}}(w_s)\right\|_1 (1 - \sqrt{\epsilon}). \tag{23}$$

Let $\sigma_s$ be the vector such that $\sigma_{sj} := \frac{(\nabla\widehat{\mathcal{R}}(w_s))_j}{\left|(\nabla\widehat{\mathcal{R}}(w_s))_j\right| + \epsilon}$. Then, by the preceding inequality,

$$\langle Z\sigma_s, q_s \rangle = \frac{1}{-\ell'(\psi(p_s))} \sum_{j \in [d]} \frac{(\nabla\widehat{\mathcal{R}}(w_s))_j}{\left|(\nabla\widehat{\mathcal{R}}(w_s))_j\right| + \epsilon}$$

$$\geq \frac{\left\|\nabla\widehat{\mathcal{R}}(w_s)\right\|_1}{-\ell'(\psi(p_s))}(1 - \sqrt{\epsilon})$$

$$= \|Z^\intercal q_s\|_1 (1 - \sqrt{\epsilon}).$$

Therefore, by Lemma 18, Assumption 1, and let $\kappa := \frac{12C^2}{\gamma\left(1-\sqrt{\beta_2}\right)^2}$,

$$\langle Zg_s, q_s \rangle = \langle Z\sigma_s, q_s \rangle - \langle Zg_s - Z\sigma_s, q_s \rangle$$

$$\geq \|Z^\intercal q_s\|_1 (1 - \sqrt{\epsilon}) - \frac{1}{\ell'(\psi(Zw_s))} \frac{12C^2}{\left(1-\sqrt{\beta_2}\right)^2} \eta\widehat{\mathcal{G}}(w_s)$$

$$\geq \|Z^\intercal q_s\|_1 (1 - \sqrt{\epsilon}) - \frac{1}{\ell'(\psi(Zw_s))} \frac{12C^2}{\gamma\left(1-\sqrt{\beta_2}\right)^2} \eta\left\|\nabla\widehat{\mathcal{R}}(w_s)\right\|$$

$$= \|Z^\intercal q_s\|_1 \left(1 - \kappa\eta - \sqrt{\epsilon}\right).$$

Plugging the preceding inequality to eq. (22) and since $-\psi(Zw_0) \geq -n$,

$$\psi(Zw_t) \geq -n + \sum_{s=0}^{t} \eta \left(1 - \eta\kappa - \sqrt{\epsilon}\right)\|Z^\mathsf{T} q_s\| - \eta^2 n\|Zd_s\|_\infty^2$$

$$\geq \sum_{s=0}^{t} \eta\gamma \left(1 - \eta\kappa - \sqrt{\epsilon} - \frac{\eta n C^2}{\gamma} - \frac{n}{\gamma\eta t}\right). \tag{24}$$

Since $\eta = \frac{1}{\sqrt{t}}$, time $t$ satisfies $t \geq \frac{4096C^4 n^2}{\gamma^2(1-\sqrt{\beta_2})^2}$, and $\sqrt{\epsilon} \leq \frac{1}{4}$, each summand is positive.

Therefore, to get a margin lower bound from here, it suffices to divide both sides of eq. (24) by $\|w_t\|_\infty$ and upper bound $\|w_t\|_\infty$. By triangle inequality and since $\|g_s\|_\infty \leq C$ by Lemma 7,

$$\|w_t\|_\infty \leq \sum_{s<t} \eta\|g_s\|_\infty \leq C\eta t.$$

Hence, by the preceding inequality and eq. (24),

$$\frac{\psi(Zw_t)}{\|w_t\|_\infty} \geq \frac{1}{C\eta t} \sum_{s<t} \eta\gamma \left(1 - \eta\kappa - \sqrt{\epsilon} - \frac{\eta n C^2}{\gamma} - \frac{n}{\gamma\eta t}\right)$$

$$= \frac{\gamma}{C} \left(1 - \eta\kappa - \sqrt{\epsilon} - \frac{\eta n C^2}{\gamma} - \frac{n}{\gamma\eta t}\right)$$

$$= \frac{\gamma}{C} \left(1 - \sqrt{\epsilon}\right) - \frac{\eta}{C} \left(\gamma\kappa + nC^2 + \frac{n}{\eta^2 t}\right)$$

$$= \frac{\gamma}{C} \left(1 - \sqrt{\epsilon}\right) - \frac{\eta}{C} \left(\gamma\kappa + nC^2 + n\right)$$

$$\geq \frac{\gamma}{C} \left(1 - \sqrt{\epsilon}\right) - \frac{24C^2 n}{(1-\sqrt{\beta_2})^2\sqrt{t}}.$$

$\square$

# E   STOCHASTIC ADAM

## E.1   CONCENTRATION INEQUALITIES

In this section, we collect various concentration inequalities needed for the proof of Theorem 34. We open with a lemma copied essentially verbatim from Beygelzimer et al. (2011, Theorem 1) that provides a Freedman type concentration inequality for martingales.

**Lemma 30.** *Let $X_1, \ldots, X_t$ be a sequence of real-valued random variables such that $X_t \leq R$ almost surely for some constant $R > 0$. Abbreviating $\mathbb{E}_t[X] := \mathbb{E}[X \mid X_1, \ldots, X_{t-1}]$, assume the conditional mean $\mathbb{E}_t[X_t] \leq 0$. Define the random variables,*

$$S = \sum_{t=1}^{T} X_t, \quad V = \sum_{t=1}^{T} \mathbb{E}_t[X_t^2].$$

*Then for any $\delta > 0$, with probability at least $1 - \delta$, and $\lambda \in [0, \frac{1}{R}]$, we have*

$$S \leq (e-2)\lambda V + \frac{\ln(1/\delta)}{\lambda}. \tag{25}$$

*Proof.* For any fixed $\lambda \in [0, \frac{1}{R}]$, applying the inequality $e^z \leq 1 + z + (e-2)z^2$ for any $z \leq 1$ and recalling that $\lambda X_t \leq 1$ almost surely, $\mathbb{E}_t[X_t] \leq 0$,

$$
\begin{aligned}
\mathbb{E}_t\left[\exp\left(\lambda X_t\right)\right] &\leq \mathbb{E}_t\left[1 + \lambda X_t + (e-2)\lambda^2 X_t^2\right] \\
&\leq 1 + \lambda \mathbb{E}_t[X_t] + (e-2)\lambda^2 \mathbb{E}_t\left[X_t^2\right] \\
&\leq 1 + (e-2)\lambda^2 \mathbb{E}_t\left[X_t^2\right] \\
&\leq \exp\left((e-2)\lambda^2 \mathbb{E}_t\left[X_t^2\right]\right).
\end{aligned}
\tag{26}
$$

Defining the random variables $Z_0 = 1$ and for $t \geq 1$,

$$
Z_t = Z_{t-1} \cdot \exp\left(\lambda X_t - (e-2)\lambda^2 \mathbb{E}_t\left[X_t^2\right]\right).
$$

By eq. (26),

$$
\mathbb{E}_t\left[Z_t\right] = Z_{t-1} \cdot \exp\left(\lambda X_t\right) \cdot \exp\left(-(e-2)\lambda^2 \mathbb{E}_t\left[X_t^2\right]\right) \leq Z_{t-1},
$$

which implies,

$$
\mathbb{E}[Z_t] = \mathbb{E}\mathbb{E}_t[Z_t] \leq \mathbb{E}[Z_{t-1}],
$$

and hence,

$$
\mathbb{E}[Z_T] \leq \mathbb{E}[Z_{T-1}] \leq \cdots \leq \mathbb{E}[Z_0] = 1.
$$

Applying Markov's inequality to $Z_T$ and by the preceding inequality,

$$
\Pr\left(Z_T \geq \frac{1}{\delta}\right) \leq \delta.
$$

Recalling that $Z_T = \exp\left(\lambda S - (e-2)\lambda^2 V\right)$ and using the preceding inequality gives the desired result. $\qquad\square$

**Lemma 31.** *Under the same assumptions as Theorem 34, with probability at least $1 - \delta$,*

$$
\left\|\sum_{i=1}^N \frac{\ell'(\langle w_s, \overline{z}_i\rangle)}{N}\overline{z}_i - \mathbb{E}_z\left[\nabla\ell(\langle w_s, z\rangle)\right]\right\|_1 \leq d\sqrt{\frac{2\ln(\frac{2td}{\delta})}{N}},
$$

*and, with probability at least $1 - \delta$,*

$$
\left|\sum_{i=1}^N \frac{\ell'(\langle w_s, \overline{z}_i\rangle)}{N} - \mathbb{E}_z\left[\ell'\left(\langle w_s, \overline{z}_i\rangle\right)\right]\right| \leq \sqrt{\frac{\ln(\frac{2t}{\delta})}{2N}}.
$$

*Proof.* Fix $j \in [d]$ and time $s \leq t$. Let $X_i$ be the random variable such that

$$
X_i := \ell'(\langle w_s, \overline{z}_i\rangle)\overline{z}_{ij}.
$$

Since $|X_i| \leq 1$, by Hoeffding's inequality, with probability at least $1 - \delta$,

$$
\left|\sum_{i=1}^N \frac{\ell'(\langle w_s, \overline{z}_i\rangle)}{N}\overline{z}_{ij} - \mathbb{E}_z\left[\ell'(\langle w_s, z\rangle)\overline{z}_j\right]\right| \leq \sqrt{\frac{2\ln(\frac{2}{\delta})}{N}}.
$$

Union bounding over coordinates $j \in [d]$ and time $s \leq t$, with probability at least $1 - \delta$,

$$
\left\|\sum_{i=1}^N \frac{\ell'(\langle w_s, \overline{z}_i\rangle)}{N}\overline{z}_i - \mathbb{E}_z\left[\nabla\ell(\langle w_s, z\rangle)\right]\right\|_1 \leq d\sqrt{\frac{2\ln(\frac{2td}{\delta})}{N}}.
$$

Similarly, fix time $s < t$. For $i \in [N]$, let $Y_i$ be the random variable such that
$$Y_i := \ell'(\langle w_s, \overline{z}_i \rangle).$$

Since $X_i \in (-1, 0)$, by Hoeffding's inequality, with probability at least $1 - \delta$,
$$\left| \sum_{i=1}^{N} \frac{\ell'(\langle w_s, \overline{z}_i \rangle)}{N} - \mathbb{E}_z \left[ \ell'\left(\langle w_s, \overline{z}_i \rangle\right) \right] \right| \le \sqrt{\frac{\ln(\frac{2}{\delta})}{2N}}.$$

Union bounding over time $s < t$, with probability at least $1 - \delta$,
$$\left| \sum_{i=1}^{N} \frac{\ell'(\langle w_s, \overline{z}_i \rangle)}{N} - \mathbb{E}_z \left[ \ell'\left(\langle w_s, \overline{z}_i \rangle\right) \right] \right| \le \sqrt{\frac{\ln(\frac{2t}{\delta})}{2N}}.$$

$\square$

The following lemma controls the squared gradient term.

**Lemma 32.** *Suppose $\eta_s = \eta \le \frac{(1-\beta_1)^2}{4CR^2}$ where $C := \frac{1}{\sqrt{1-\beta_1}}$ and the iterates $w_s$ are updated using eq. (1). Then there exists a time $k < t$ such that*
$$G(w_k) \le \frac{2048 \ln(2\sqrt{n}d/\delta)}{\gamma^2 \sqrt{n}},$$
*or, with probability at least $1 - \delta$, for all time $s < t$,*
$$\left\| \overline{Z} g_s \right\|_\infty^2 \le \frac{\gamma^2}{256\epsilon^2} G(w_s)^2. \tag{27}$$

*Proof.* Define the set $I := \{(\tau, b) \, | \, 1 \le \tau \le t \text{ and } b \in [B]\}$. Let $i : \mathbb{N} \times \mathbb{N} \to \mathbb{N}$ be the function $i(\tau, b) := \tau B + b$. Note that $i$ induces a total order on $I$. For the remainder of the proof, identify the integer $i(\tau, b)$ with the tuple $(\tau, b)$. In particular, for any random variable $X_{(\tau,b)}$ abbreviate the conditional mean as
$$\mathbb{E}_{i(\tau,b)} \left[ X_{(\tau,b)} \right] := \mathbb{E} \left[ X_{(\tau,b)} \mid X_{(r,c)} \text{ s.t } (r,c) \in I, \, i(r,c) < i(\tau,b) \right].$$

Fix time $s$ and $j \in [d]$. Further pick time $\tau \le s$ and data $z_b \in B_\tau$, and define the random variable,
$$X_{(\tau,b)} := a_{s,\tau} \left( \ell'(\langle w_\tau, z_b \rangle) - \mathbb{E} \left[ \ell'(\langle w_s, z \rangle) \right] \right).$$

Since $w_\tau$ is independent of data $(x, y) \in B_\tau$, elements of $B_\tau$ are pairwise independent, and $a_{s,\tau}$ is a deterministic constant, the conditional expectation satisfies $\mathbb{E}_{i(\tau,b)}[X_{(\tau,b)}] = 0$. Hence $X_{(\tau,b)}$ is a martingale difference sequence.

To obtain an almost sure bound on $X_{(\tau,b)}$, note that $\ell' \in (-1, 0)$, and $\{a_{s,\tau}\}_{\tau=0}^{s}$ is a nondecreasing positive sequence. Hence, $\left| X_{(\tau,b)} \right| \le a_{s,s} \le 1$.

Furthermore, the conditional variance satisfies,
$$\mathbb{E}_{i(\tau,b)}[X_{(\tau,b)}^2] = \mathbb{E}_z[a_{s,\tau}^2 \ell'(\langle w_s, z \rangle)^2] - \mathbb{E}_z \left[ a_{s,\tau} \ell'(\langle w_s, z \rangle) \right]^2 \le a_{s,\tau}^2 \mathbb{E}_z[\ell'(\langle w_s, z \rangle)^2].$$

By definition of $\hat{G}, G$, it follows that
$$\sum_{k \le s} a_{s,k} \left( \hat{G}_k(w_k) - G(w_k) \right) = \sum_{\ell \in I} \frac{X_\ell}{B}.$$

Hence, applying Theorem 30 with $\lambda = \frac{\gamma}{64}$, and union bounding over $s < t$, with probability at least $1 - \delta$, for all $j \in [d]$ and time $s < t$,
$$\sum_{k \le s} a_{s,k} \left( \hat{G}_k(w_k) - G(w_k) \right) \le \frac{64 \ln(2td/\delta)}{\gamma B} + \frac{\gamma}{64} \sum_{k \le s} a_{s,k}^2 \frac{\mathbb{E} \left[ \ell'(\langle w_k, z \rangle)^2 \right]}{B}.$$

Since $\left|\ell'\right| \leq 1$ and $a_{s,k} \leq 1$, by the definition of $G$ and Lemma 16, and since $\eta \leq \frac{(1-\beta_1)^2}{4CR^2}$, the preceding inequality simplifies,

$$\sum_{k \leq s} a_{s,k}\left(\hat{G}_k(w_k) - G(w_k)\right) \leq \frac{64 \ln(2td/\delta)}{\gamma B} + \frac{\gamma}{64} \sum_{k \leq s} a_{s,k}^2 G(w_k)$$

$$\leq \frac{64 \ln(2td/\delta)}{\gamma B} + \frac{\gamma}{64} \sum_{k \leq s} a_{s,k} G(w_k)$$

$$\leq \frac{64 \ln(2td/\delta)}{\gamma B} + \frac{\gamma}{64}\left(1 + \frac{4CR^2}{(1-\beta_1)^2}\eta\right)G(w_s)$$

$$\leq \frac{64 \ln(2td/\delta)}{\gamma B} + \frac{\gamma}{32} G(w_s). \tag{28}$$

Consider two cases.

1. Suppose there exists time $k < t$ such that

$$\frac{\gamma}{32} G(w_k) \leq \frac{64 \ln(2td/\delta)}{\gamma B}.$$

Since $B = t = \sqrt{n}$,

$$G(w_k) \leq \frac{2048 \ln(2\sqrt{n}d/\delta)}{\gamma^2 \sqrt{n}}.$$

2. Suppose for all time $s < t$,

$$\frac{\gamma}{32} G(w_k) > \frac{64 \ln(2td/\delta)}{\gamma B}.$$

Then by eq. (28),

$$\sum_{k \leq s} a_{s,k}\left(\hat{G}_k(w_k) - G(w_k)\right) \leq \frac{\gamma}{16} G(w_s).$$

Therefore, since $\|m_s\|_2, \|z_s\|_2 \leq 1$ and by the preceding inequality and triangle inequality,

$$\left\|\overline{Z}g_s\right\|_\infty^2 = \max_{i \in [N]} \left(\overline{z}_i^\mathsf{T} g_s\right)^2$$

$$\leq \max_{i \in [N]} \left(\sum_{j \in [d]} \overline{z}_{ij} \frac{m_{sj}}{\sqrt{v_{sj}} + \epsilon}\right)^2$$

$$\leq \frac{1}{\epsilon^2} \max_{i \in [N]} \|z_i\|_2^2 \|m_s\|_2^2$$

$$\leq \frac{1}{\epsilon^2} \|m_s\|_2^2$$

$$\leq \frac{1}{\epsilon^2} \left(\sum_{k \leq s} a_{s,k} \hat{G}_s(w_s)\right)^2$$

$$\leq \frac{\gamma^2}{256\epsilon^2} G(w_s)^2. \tag{29}$$

$\square$

We define the averaged first moment of the gradients as $M_s := \sum_{k \leq s} a_{s,k} \mathbb{E}_{z \sim \mathcal{D}}\left[\nabla \ell \langle w_s, z \rangle\right]$ and abbreviate the expected gradient at time $s$ as $E_s := \mathbb{E}_{z \sim \mathcal{D}}\left[\nabla \ell \left(\langle w_s, z \rangle\right)\right]$. The following lemma controls the deviation between the estimate of the averaged first moment and the true averaged first moment of the gradients.

**Lemma 33.** *Suppose $\eta_s = \eta \le \frac{(1-\beta_1)^2}{4CR^2}$ where $C := \frac{1}{\sqrt{1-\beta_1}}$ and the iterates $w_s$ are updated using eq. (1). Then there exists a time $k < t$ such that*

$$G(w_k) \le \frac{128d\ln(2\sqrt{n}d/\delta)}{\epsilon^2\gamma^2\sqrt{n}},$$

*or, with probability at least $1 - \delta$, for all time $s < t$,*

$$\left| \left\langle \frac{m_s - M_s}{\sqrt{v_s} + \epsilon}, \mathbb{E}\left[\nabla\ell(w_k)\right] \right\rangle \right| \le \frac{\gamma}{4}G(w_s)\|E_s\|_2. \tag{30}$$

*Proof.* Define the set $I := \{(\tau, b) \,|\, 1 \le \tau \le t \text{ and } b \in [B]\}$. Let $i : \mathbb{N} \times \mathbb{N} \to \mathbb{N}$ be the function $i(\tau, b) := \tau B + b$. Note that $i$ induces a total order on $I$. For the remainder of the proof, identify the integer $i(\tau, b)$ with the tuple $(\tau, b)$. In particular, for any random variable $X_{(\tau,b)}$ abbreviate the conditional mean as

$$\mathbb{E}_{i(\tau,b)}\left[X_{(\tau,b)}\right] := \mathbb{E}\left[X_{(\tau,b)} \mid X_{(r,c)} \text{ s.t } (r, c) \in I \,, i(r, c) < i(\tau, b)\right].$$

Fix time $s$ and $j \in [d]$. Further pick time $\tau \le s$ and data $z_b \in B_\tau$, and define the random variable,

$$X_{(\tau,b)} := a_{s,\tau}\left(\ell'(\langle w_\tau, z_b \rangle)z_{b,j} - \mathbb{E}\left[\ell'(\langle w_s, z \rangle)z_j\right]\right).$$

Since $w_\tau$ is independent of data $(x, y) \in B_\tau$, elements of $B_\tau$ are pairwise independent, and $a_{s,\tau}$ is a deterministic constant, the conditional expectation satisfies $\mathbb{E}_{i(\tau,b)}[X_{(\tau,b)}] = 0$. Hence $X_{(\tau,b)}$ is a martingale difference sequence.

To obtain an almost sure bound on $X_{(\tau,b)}$, note that $|\ell'| \le 1, |z_j| \le 1$ for any $z \sim \mathcal{D}$ almost surely, and $\{a_{s,\tau}\}_{\tau=0}^s$ is a monotonically increasing positive sequence, $|X_{(\tau,b)}| \le 2a_{s,s}$.

Furthermore, the conditional variance satisfies,

$$\mathbb{E}_{i(\tau,b)}[X_{(\tau,b)}^2] = \mathbb{E}_z[a_{s,\tau}^2\ell'(\langle w_s, z \rangle)^2 z_j^2] - \mathbb{E}_z\left[a_{s,\tau}\ell'(\langle w_s, z \rangle)z_j\right]^2 \le a_{s,\tau}^2\mathbb{E}_z[\ell'(\langle w_s, z \rangle)^2 z_j^2].$$

By definition of $m_s, M_s$, it follows that

$$m_{sj} - M_{sj} = \sum_{\ell \in I} \frac{X_\ell}{B}.$$

Hence, applying Theorem 30 with $\lambda = \frac{\gamma\epsilon}{16\sqrt{d}}$, and union bounding over $j \in [d]$ and $s < t$, with probability at least $1 - \delta$, for all $j \in [d]$ and time $s < t$,

$$|m_{sj} - M_{sj}| \le \frac{16\sqrt{d}\ln(2td/\delta)}{\epsilon\gamma B} + \frac{\gamma\epsilon}{16\sqrt{d}}\sum_{k \le s} a_{s,k}^2 \frac{\mathbb{E}\left[\ell'(\langle w_k, z \rangle)^2 z_j^2\right]}{B}. \tag{31}$$

By the preceding inequality and since $\|\cdot\|_1 \leq \sqrt{d}\|\cdot\|_2$,

$$
\begin{aligned}
\left| \left\langle \frac{m_s - M_s}{\sqrt{v_s} + \epsilon}, E_s \right\rangle \right| &= \left| \sum_{j \in [d]} \frac{(m_{sj} - M_{sj}) E_{sj}}{\sqrt{v_{sj}} + \epsilon} \right| \\
&\leq \frac{1}{\epsilon} \sum_{j \in [d]} |m_{sj} - M_{sj}| |E_{sj}| \\
&\leq \sum_{j \in [d]} \frac{16\sqrt{d} \ln(2td/\delta)}{\epsilon^2 \gamma B} |E_{sj}| + \frac{\gamma}{16\sqrt{d}} \sum_{k \leq s} a_{s,k}^2 \frac{\mathbb{E}\left[\ell'(\langle w_k, z \rangle)^2 z_j^2\right]}{B} |E_{sj}| \\
&= \frac{16\sqrt{d} \ln(2td/\delta)}{\epsilon^2 \gamma B} \|E_s\|_1 + \sum_{j \in [d]} \frac{\gamma}{16\sqrt{d}} \sum_{k \leq s} a_{s,k}^2 \frac{\mathbb{E}\left[\ell'(\langle w_k, z \rangle)^2 z_j^2\right]}{B} |E_{sj}| \\
&\leq \frac{16d \ln(2td/\delta)}{\epsilon^2 \gamma B} \|E_s\|_2 + \sum_{j \in [d]} \frac{\gamma}{16\sqrt{d}} \sum_{k \leq s} a_{s,k}^2 \frac{\mathbb{E}\left[\ell'(\langle w_k, z \rangle)^2 z_j^2\right]}{B} |E_{sj}|.
\end{aligned}
\tag{32}
$$

Pushing the summation over the coordinates $j \in [d]$ inside, applying Cauchy-Schwarz, Jensen's inequality, and since $|\ell'| \leq 1$ and for any $\|z\|_2 \leq 1$ almost surely, the second term in eq. (32) can be bounded as follows,

$$
\begin{aligned}
\sum_{j \in [d]} \frac{\gamma}{16\sqrt{d}} \sum_{k \leq s} a_{s,k}^2 \frac{\mathbb{E}\left[\ell'(\langle w_k, z \rangle)^2 z_j^2\right]}{B} |E_{sj}| &= \frac{\gamma}{16\sqrt{d}} \sum_{k \leq s} a_{s,k}^2 \sum_{j \in [d]} \frac{\mathbb{E}\left[\ell'(\langle w_k, z \rangle)^2 z_j^2\right]}{B} |E_{sj}| \\
&\leq \frac{\gamma}{16\sqrt{d}} \sum_{k \leq s} a_{s,k}^2 \frac{\sqrt{d}}{B} \sum_{j \in [d]} \mathbb{E}\left[\ell'(\langle w_k, z \rangle)^2 z_j^2\right] \|E_s\|_2 \\
&\leq \frac{\gamma}{16\sqrt{d}} \sum_{k \leq s} a_{s,k}^2 \frac{\sqrt{d}}{B} \mathbb{E}\left[\ell'(\langle w_k, z \rangle)^2 \|z\|_2^2\right] \|E_s\|_2 \\
&\leq \frac{\gamma}{16\sqrt{d}} \sum_{k \leq s} a_{s,k}^2 \frac{\sqrt{d}}{B} \mathbb{E}\left[-\ell'(\langle w_k, z \rangle)\right] \|E_s\|_2 \\
&= \frac{\gamma}{16} \|E_s\|_2 \sum_{k \leq s} a_{s,k}^2 G(w_k) \\
&\leq \frac{\gamma}{16} \|E_s\|_2 \sum_{k \leq s} a_{s,k} G(w_k).
\end{aligned}
$$

Applying Lemma 16 to the preceding inequality and since $\eta \leq \frac{(1-\beta_1)^2}{4CR^2}$,

$$
\begin{aligned}
\sum_{j \in [d]} \frac{\gamma}{16\sqrt{d}} \sum_{k \leq s} a_{s,k}^2 \frac{\mathbb{E}\left[\ell'(\langle w_k, z \rangle)^2 z_j^2\right]}{B} |E_{sj}| &\leq \frac{\gamma}{16} \|E_s\|_2 \sum_{k \leq s} a_{s,k} G(w_k) \\
&\leq \frac{\gamma}{16} \left(1 + \frac{4CR^2}{(1-\beta_1)^2} \eta\right) G(w_s) \|E_s\|_2 \\
&\leq \frac{\gamma}{8} G(w_s) \|E_s\|_2.
\end{aligned}
\tag{33}
$$

Applying eq. (33) to eq. (32),

$$
\left| \left\langle \frac{m_s - M_s}{\sqrt{v_s} + \epsilon}, E_s \right\rangle \right| \leq \left( \frac{16d \ln(2td/\delta)}{\epsilon^2 \gamma B} + \frac{\gamma}{8} G(w_s) \right) \|E_s\|_2.
\tag{34}
$$

Now consider two cases.

1. Suppose for some time $k < t$

$$\frac{\gamma}{8} G(w_k) \leq \frac{16d \ln(2td/\delta)}{\epsilon^2 \gamma B}.$$

Since $B = t = \sqrt{n}$, dividing both sides by $\frac{\gamma}{8}$,

$$G(w_k) \leq \frac{128d \ln(2\sqrt{n}d/\delta)}{\epsilon^2 \gamma^2 \sqrt{n}}.$$

2. Suppose for all time $s < t$,

$$\frac{\gamma}{8} G(w_k) > \frac{16d \ln(2td/\delta)}{\epsilon^2 \gamma B}.$$

Then by eq. (34),

$$\left| \left\langle \frac{m_s - M_s}{\sqrt{v_s} + \epsilon}, E_s \right\rangle \right| \leq \left( \frac{16d \ln(2td/\delta)}{\epsilon^2 \gamma B} + \frac{\gamma}{8} G(w_s) \right) \|E_s\|_2 \leq \frac{\gamma}{4} G(w_s) \|E_s\|_2.$$

$\square$

### E.2 Population Loss Minimization

In this section, we prove Theorem 34.

**Theorem 34.** *Assume the distributional linear separability: Assumption 2, with $\gamma = \gamma_2$. Further assume, for $z = yx \sim \mathcal{D}$, $\|z\|_2 \leq 1$ and $\|z\|_1 \leq R$ almost surely. For any $0 \leq \beta_1 \leq \beta_2 < 1$ and $0 < \epsilon \leq 1$, with probability at least $1 - 10\delta$, for constant learning rate $\eta = \frac{\epsilon^2 (1 - \beta_1)^{5/2} \gamma_2}{4R^2}$, batch size $B = \sqrt{n}$, and iteration count $t = \sqrt{n}$,*

$$\min_{s<t} \Pr(\langle w_s, z \rangle < 0) \leq \frac{8192R^2}{(1 - \beta_1)^{5/2} \epsilon^2 \gamma_2^3 \sqrt{n}}. \tag{35}$$

*Proof.* By Lemma 24, we have the following regret guarantee,

$$\overline{\mathcal{R}}(w_0) - \overline{\mathcal{R}}(w_t) \geq \sum_{s<t} \eta \left( \left\langle g_s, \overline{Z}^\intercal \bar{q}_s \right\rangle - \eta \overline{G}(w_s) \left\| \overline{Z} g_s \right\|_\infty^2 \right). \tag{36}$$

Excluding $\delta$ failure probability from invoking Theorem 31 and by Lemma 7, for all time $s < t$,

$$\left\langle g_s, \overline{Z}^\intercal \bar{q}_s \right\rangle = \left\langle g_s, \mathbb{E}_z \left[ \nabla \ell \left( \langle w_s, z \rangle \right) \right] \right\rangle + \left\langle g_s, \overline{Z}^\intercal \bar{q}_s - \mathbb{E}_z \left[ \nabla \ell \left( \langle w_s, z \rangle \right) \right] \right\rangle$$

$$\geq \left\langle g_s, \mathbb{E}_z \left[ \nabla \ell \left( \langle w_s, z \rangle \right) \right] \right\rangle - \|g_s\|_\infty \left\| \overline{Z}^\intercal \bar{q}_s - \mathbb{E}_z \left[ \nabla \ell \left( \langle w_s, z \rangle \right) \right] \right\|$$

$$\geq \left\langle g_s, \mathbb{E}_z \left[ \nabla \ell \left( \langle w_s, z \rangle \right) \right] \right\rangle - Cd \sqrt{\frac{2 \ln(\frac{2td}{\delta})}{N}}. \tag{37}$$

Similarly, excluding another $\delta$ failure probability, by Lemma 7 and Theorem 31, for all time $s < t$,

$$\overline{G}(w_s) \left\| \overline{Z} g_s \right\|^2 \leq G(w_s) \left\| \overline{Z} g_s \right\|_\infty^2 + \sqrt{\frac{\ln(\frac{2t}{\delta})}{2N}} \left\| \overline{Z} g_s \right\|_\infty^2 \leq G(w_s) \left\| \overline{Z} g_s \right\|_\infty^2 + C^2 R^2 \sqrt{\frac{\ln(\frac{2t}{\delta})}{2N}}. \tag{38}$$

Applying eqs. (37) and (38) to eq. (36) and rearranging terms,

$$\overline{\mathcal{R}}(w_0) - \overline{\mathcal{R}}(w_t) + 4\eta t C^2 R^2 d \sqrt{\frac{2 \ln(\frac{td}{\delta})}{N}} \geq \sum_{s<t} \eta \left( \left\langle g_s, \mathbb{E}_z \left[ \nabla \ell \left( \langle w_s, z \rangle \right) \right] \right\rangle - \eta G(w_s) \left\| \overline{Z} g_s \right\|_\infty^2 \right).$$

Recall that $N$ was the number of *unseen* data. Taking $N \geq 32 \left(\eta t C^2 R^2 d\right)^2 \ln(\frac{td}{\delta})$, we can simplify the mirror descent guarantee,

$$\overline{\mathcal{R}}(w_0) - \overline{\mathcal{R}}(w_t) + 1 \geq \sum_{s<t} \eta \left( \left\langle g_s, \mathbb{E}_{z \sim \mathcal{D}} \left[ \nabla \ell \left( \langle w_s, z \rangle \right) \right] \right\rangle - \eta G(w_s) \left\| \overline{Z} g_s \right\|_\infty^2 \right). \tag{39}$$

We first handle the inner product term. Recall that we have defined $M_s = \sum_{k \leq s} a_{s,k} \mathbb{E}_{z \sim \mathcal{D}} \left[ \nabla \ell \langle w_s, z \rangle \right]$ and $E_s := \mathbb{E}_{z \sim \mathcal{D}} \left[ \nabla \ell \left( \langle w_s, z \rangle \right) \right]$. Adding and subtracting various quantities,

$$\langle g_s, E_s \rangle = \left\langle \frac{E_s}{\sqrt{v_s} + \epsilon}, E_s \right\rangle + \left\langle \frac{M_s - E_s}{\sqrt{v_s} + \epsilon}, E_s \right\rangle + \left\langle g_s - \frac{M_s}{\sqrt{v_s} + \epsilon}, E_s \right\rangle. \tag{40}$$

We tackle each term separately. We start by handling the first term. By Assumption 2,

$$\|E_s\|_2 = \left\| \mathbb{E}_{z \sim \mathcal{D}} \left[ \nabla \ell \left( \langle w_s, z \rangle \right) \right] \right\|_2 = \sup_{\|u\|_2 \leq 1} \left\langle u, \mathbb{E}_{z \sim \mathcal{D}} \left[ \nabla \ell \left( \langle w_s, z \rangle \right) \right] \right\rangle \geq \gamma_2 G(w_s).$$

Since $\sqrt{v_{sj}} \leq 1$ for all $j \in [d]$ and $\epsilon \leq 1$, by the preceding inequality,

$$\left\langle \frac{E_s}{\sqrt{v_s} + \epsilon}, E_s \right\rangle = \sum_{j \in d} \frac{E_{sj}^2}{\sqrt{v_{sj}} + \epsilon} \geq \frac{1}{1+\epsilon} \|E_s\|_2^2 \geq \frac{\gamma_2}{1+\epsilon} G(w_s) \|E_s\|_2 \geq \frac{\gamma_2}{2} G(w_s) \|E_s\|_2. \tag{41}$$

We now handle the second term in eq. (40). Applying Lemma 16 with $\beta = \beta_1$ and letting $C = \frac{1}{\sqrt{1-\beta_1}}$ be the constant defined in Lemma 16,

$$\|M_s - E_s\|_2 = \left\| \sum_{k \leq s} a_{s,k} \mathbb{E}_z \left[ \nabla \ell \left( \langle w_k, z \rangle \right) \right] - \mathbb{E}_z \left[ \nabla \ell \left( \langle w_s, z \rangle \right) \right] \right\|_2 \leq \eta \frac{4 C R^2}{(1-\beta_1)^2} G(w_s).$$

Hence, by the preceding inequality and since $\eta = \frac{\gamma_2 (1-\beta_1)^2 \epsilon}{32 C R^2}$,

$$\left\langle \frac{M_s - E_s}{\sqrt{v_s} + \epsilon}, E_s \right\rangle \leq \frac{1}{\epsilon} \|M_s - E_s\|_2 \|E_s\|_2 \tag{42}$$

$$\leq \eta \left( \frac{4 C R^2}{\epsilon (1-\beta_1)^2} \right) G(w_s) \|E_s\|_2$$

$$\leq \frac{\gamma_2}{8} G(w_s) \|E_s\|_2. \tag{43}$$

Now we handle the third term in eq. (40). By Lemma 33, either there exists a time $k < t$ such that

$$G(w_k) \leq \frac{128 d \ln(2\sqrt{n} d/\delta)}{\epsilon^2 \gamma_2^2 \sqrt{n}},$$

or, with probability at least $1 - \delta$, for all time $s < t$,

$$\left| \left\langle \frac{m_s - M_s}{\sqrt{v_s} + \epsilon}, \mathbb{E} \left[ \nabla \ell(w_k) \right] \right\rangle \right| \leq \frac{\gamma_2}{4} G(w_s) \|E_s\|_2. \tag{44}$$

In the former case, we are done. Hence, suppose we are in the latter case. Then,

$$\left\langle g_s - \frac{M_s}{\sqrt{v_s} + \epsilon}, E_s \right\rangle = \left\langle \frac{m_s - M_s}{\sqrt{v_s} + \epsilon}, E_s \right\rangle \leq \frac{\gamma_2}{4} G(w_s) \|E_s\|_2 \leq \frac{\gamma_2}{4} G(w_s) \|E_s\|_2. \tag{45}$$

Therefore, by applying eqs. (40) to (42) and (45) to eq. (39) and by the definition of $\eta, \beta$,

$$\overline{\mathcal{R}}(w_0) - \overline{\mathcal{R}}(w_t) + 1 \geq \sum_{s<t} \eta \left( \frac{\gamma_2}{8} G(w_s) \|E_s\|_2 - \eta G(w_s) \left\| \overline{Z}\overline{g}_s \right\|_\infty^2 \right). \tag{46}$$

It remains to control the squared gradient term. By Theorem 32, either there exists a time $k < t$ such that

$$G(w_k) \leq \frac{2048 \ln(2\sqrt{n}d/\delta)}{\gamma_2^2 \sqrt{n}},$$

or, with probability at least $1 - \delta$, for all time $s < t$,

$$\left\| \overline{Z}g_s \right\|_\infty^2 \leq \frac{\gamma^2}{256\epsilon^2} G(w_s)^2. \tag{47}$$

In the former case, we are done. Thus, suppose the latter case. Then, applying eq. (47) to eq. (46),

$$\overline{\mathcal{R}}(w_0) - \overline{\mathcal{R}}(w_t) + 1 \geq \sum_{s<t} \eta \left( \frac{\gamma_2}{8} G(w_s) \|E_s\|_2 - \eta \frac{\gamma_2^2}{256\epsilon^2} G(w_s)^3 \right).$$

Since $\|E_s\|_2 \geq \gamma_2 G(w_s)$, $\eta \leq \frac{\epsilon^2(1-\beta_1)^2}{4CR^2} \leq \epsilon^2$, and $G(w_s) \leq 1$, the preceding inequality simplifies,

$$\overline{\mathcal{R}}(w_0) - \overline{\mathcal{R}}(w_t) + 1 \geq \sum_{s<t} \eta \left( \frac{\gamma_2^2}{8} G(w_s)^2 - \frac{\gamma_2^2}{256} G(w_s)^2 \right) \geq \sum_{s<t} \eta \frac{\gamma_2^2}{16} G(w_s)^2 \geq \frac{\eta t \gamma_2^2}{16} \inf_{s<t} G(w_s)^2.$$

Dividing $\frac{\eta t \gamma_2^2}{16}$ on both sides grants and recalling that $w_0 = 0$ which implies $\overline{\mathcal{R}}(w_0) = \ln(2) < 1$,

$$\frac{32}{\eta \gamma_2^2 t} \geq \frac{16}{\eta \gamma_2^2 t} \left( \overline{\mathcal{R}}(w_0) - \overline{\mathcal{R}}(w_t) + 1 \right) \geq \inf_{s \leq t} G(w_s)^2.$$

Furthermore,

$$\Pr(\langle w_s, z \rangle < 0) \leq 4G(w_s)^2.$$

Putting everything together and since $t = \sqrt{n}$ and $\eta = \frac{\epsilon^2(1-\beta_1)^{5/2}\gamma_2}{4R^2}$,

$$\inf_{s<t} \Pr(\langle w_s, z \rangle < 0) \leq \frac{128}{\eta \gamma_2^2 t} = \frac{512R^2}{\gamma_2^3 \epsilon^2 (1-\beta_1)^{5/2} \sqrt{n}} \leq \frac{8192R^2}{\gamma_2^3 \epsilon^2 (1-\beta_1)^{5/2} \sqrt{n}}.$$

$\square$

