# OpenReview forum: "Understanding Adam through the Lens of Duality: A Unified Theory of Normalized Gradient Methods"
_ICLR.cc/2026/Conference — ICLR 2026 Conference Withdrawn Submission_

### Official Review · Reviewer_wEgr · 2025-10-14

**Soundness:** 1
**Presentation:** 1
**Contribution:** 1
**Rating:** 0
**Confidence:** 5

**Summary:**

This paper studies the implicit bias of normalized steepest descent, including the SignGD and normalized gradient descent, and generalizes their results to Adam, which shares similar properties with SignGD under some scenarios. However, all these conclusions have been rigorously established in the previous works. [1, 2, 3, 4]

[1] Gunasekar et al. Characterizing implicit bias in terms of optimization geometry. ICML 2018.

[2] Ji and Telgarsky. Characterizing the implicit bias via a primal-dual analysis. ALT 2021.

[3] Zhang et al. The implicit bias of Adam on separable data. Neurips 2024.

[4] Fan et al. Implicit bias of spectral descent and muon on multiclass separable data. Neurips 2025.

**Strengths:**

Improve one previous convergence rate.

**Weaknesses:**

As I have stated in the summary section, all the conclusions regarding the implicit bias of different optimizers in this paper have been rigorously established in the previous studies [1, 2, 3, 4]. Specifically, [1] demonstrated that under logistic regression settings, all the steepest descent w.r.t. $\\|\cdot\\|$-norm will converge to the maximum $\\|\cdot\\|$ norm, and  SignGD is exactly the normalized version of steepest descent w.r.t. the $\ell\_\\infty$. Notably, the proof of this conclusion exactly utilizes the duality between the objective norm and its dual. Given these results, [3, 4] establish the implicit bias of Adam, SignGD, and Muon by demonstrating they are normalized steepest descent w.r.t. specific norms, or share similar properties to normalized steepest descent with specific norms. **I believe these existing works have clearly and rigorously demonstrated why Adam and SignGD have the implicit bias toward the $\ell_\infty$ norm. I fail to see any novel motivation for this study, and feel surprised about the authors' claim that "Unfortunately, it is unclear from prior work why Adam exhibits a $\ell_\infty$ max margin bias'', especially given that they have cited [3, 4] in this paper.** In addition, I am also very curious why the authors could claim they established an improved convergence rate regarding the implicit bias of normalized gradient descent, where the entirely identical conclusion is demonstrated in [2], especially given that the technical framework of this paper is established upon [2] as acknowledged in Section 3, and similarly, the entirely identical convergence rate of SignGD is also established in [4].

In summary, this study does not provide any interesting novel results. More severely, authors overclaimed many existing results as their own contributions, especially given that these existing works are known to the authors.

**Questions:**

I suggest the authors consider investigating the existing works.

---

### Official Review · Reviewer_TLVt · 2025-10-23

**Soundness:** 4
**Presentation:** 2
**Contribution:** 3
**Rating:** 6
**Confidence:** 3

**Summary:**

The paper studies generalization and implicit bias of Adam, SignGD, and Normalized GD in linear classification. The analysis relies on duality. The ($\ell_\infty$) margin maximization rate that is shown for Adam is better than what was previously shown. For stochastic Adam with mini batches, they prove minimization of the population loss at a rate faster than previous work, and their result is with high probability, unlike previous work that proved an in-expactation result.

**Strengths:**

Understanding the implicit bias of Adam, SignGD and Normalized GD, as well as population loss minimization for stochastic Adam, are important questions. The paper improves the previously known bounds. Also, the analysis relies on a duality approach, which is of independent interest.

**Weaknesses:**

Overall, I think that the contribution is nice and above the acceptance threshold. But:

I have a question/concern regarding Theorem 3:
The formal theorem implies margin maximization if $\beta_1 \approx \beta_2$, while in practice we often have $\beta_1 \neq \beta_2$. In the discussion after the theorem, the authors note that the claim also holds without the $\beta_1=\beta_2$ condition, if we add an additional assumption similar to the one from Zhang et al. (2024). The assumption from Zhang et al. requires that at initialization (or at some time $t_0$) all coordinates of the gradient are bounded away from zero by some parameter $\rho$. In their paper, they remark that this assumption is mild since for all non-degenerate datasets, with probability 1, the gradient at initialization does not have zero coordinates, and the dependence of their result $\rho$ is logarithmic. Given the fact that in practice we generally have $\beta_1 \neq \beta_2$, I view their assumption as milder. So, my questions are: (1) Can the authors specify how Theorem 3 changes if we use the assumption form Zhang et al.? (2) May the authors provide a proof or a proof sketch or an explanation for how the proof changes when relying on the assumption of Zhang et al.?

Moreover, Figure 1 is unclear. As far as I can see, the authors do not explain what algorithm each curve represents. For example, what is $\ell_2$-signGD? (is it really signGD or maybe the authors meant normalized GD here?). Also, what is SignGD with non-zero epsilon? (is it full-batch Adam without momentum, as opposed to “Adam” which represents here stochastic Adam?)

Finally, I should say that I found some of the technical arguments hard to follow. More detailed derivations would make the paper easier to read.

**Questions:**

Please see the “weaknesses” section.

---

### Official Review · Reviewer_G2uE · 2025-10-28

**Soundness:** 3
**Presentation:** 2
**Contribution:** 3
**Rating:** 4
**Confidence:** 4

**Summary:**

This paper extends the duality approach and mirror descent analysis to study normalized gradient descent, signGD and Adam. With this framework, it is clear that normalized GD maximizes $\ell_2$ norm margin and signGD maximizes $\ell_\infty$ norm margin. They improve the rate of Adam maximizing $\ell_\infty$ margin with this framework. And there is a novel high probability convergence rate for Adam.

**Strengths:**

1. The duality framework clearly shows the difference between GD, normalized GD and signGD and possibly gives a unified prespective to understand any steepest descent.
2. The effect of $\epsilon$ is explicitly discussed and verified by experiments, which is often overlooked or not fully emphasized in previous works.
3. There are novel results such as the improved rate of Adam and high probability convergence rate.

**Weaknesses:**

1. It is unclear how this framework directly helps understand Adam better. In section 3, the dual objectives are clearly mentioned for GD, signGD and normalized GD but not for Adam. In section 4, the result of Adam is obtained by drawing connection to signGD. However, the connection between signGD and Adam is already mentioned in Xie and Li(2024) and Zhang et al., (2024) to help answer why Adam exhibits $\ell_\infty$-max margin bias. There are more literature on the implicit bias of steepest descent and Adam, which are given in the questions below. Therefore, this insight is not a novel contribution.
2. The theoretical results are not satisfying enough in terms of the dependence on hyperparameters, which is also admitted by the authors. The theorem 5 also requires a very large batch size $b=\sqrt{n}$, which is impractical in practice.
3. Section 5 is disconnected to the unified framework. Also it is unclear how the lower bound justifies the dependence on $d$ in the upper bound. See questions below.

**Questions:**

1. There are two important missing literatures Characterizing Implicit Bias in Terms of Optimization Geometry, Gunasekar et al. 2020 and Flavors of Margin: Implicit Bias of Steepest Descent in Homogeneous Neural Networks, Tsilivis et al. 2024. Can you compare with them?
2. I don’t understand the statement of theorem 5. $\delta$ doesn’t appear in the test error term and there is no constraint on $\delta$. If it is correct, then we can choose $\delta=0$ and the result is too good to be true. Also it relies on $\ell_2$ norm margin, which is not satisfying to me when we want to connect Adam to $\ell_\infty$ norm margin. Can you explain the intuition of using $\ell_2$ norm margin here?
3. I didn’t understand the paragraph in line 393-398 when you mention the dependence on $d$ in the test error. In theorem 5, there is no $d$ in the error term explicitly. Are you saying $R$ and $\gamma_2$ can have some dependence on $d$? Can you explain why the sample complexity is $\Omega(d)$ rather than $\Omega(d^2)$?

Typos:
1. Page 2 line 104, the test error should be $O(\frac{d}{\sqrt{n}})$.
2. In the introduction section, there are wrong references several times when you mentioned Xie and Li 2024 proved the $\ell_\infty$ max margin result while it should be Zhang et al. 2024.

---

### Official Review · Reviewer_5Mqu · 2025-11-06

**Soundness:** 2
**Presentation:** 1
**Contribution:** 2
**Rating:** 4
**Confidence:** 3

**Summary:**

This paper presents an explanation of Adam through the lens of duality and derives a unified theoretical framework. In addition, the authors consider the deterministic setting and analyze signGD and normalized GD within this framework. Furthermore, they provide a theoretical guarantee on the test error of Adam in the stochastic setting.

**Strengths:**

1. This paper provides a novel perspective for understanding Adam.
2. The paper establishes improved convergence rates compared with previous work.
3. The paper presents a unified theoretical framework that can be applied to analyze various optimization methods.

**Weaknesses:**

1. This paper is poorly written. Although I can follow the main motivation, I am still confused by many details.
   - First, the discussions on stochastic and deterministic settings should be clearly separated. For example, in Figure 1(b), the stochastic and deterministic results should not be plotted together.
   - Second, I am confused why Section 6 is titled “Future Related Work.” Normally, the related work section should appear before the preliminaries, and the authors should also discuss other papers that aim to understand Adam, such as [1] and [2].
   - Third, the authors repeatedly refer to Lemma 20 in the main text, but Lemma 20 itself only appears in the appendix.
2. In Assumption 1, $Z u$ is a vector. Does the condition $Z u > 0$ mean that all its elements are positive? If so, I suggest the authors use the notation $\succ 0$.
3. In Line 183, could the authors explain why $\mathcal{L}(Zw)=\hat{\mathcal{R}}(w)$.
4. For Adam in the online stochastic setting, could the author drive the regret bound for this problem?
5. Can the authors conduct experiments to further validate their analysis?
6. The duality framework of this paper follows previous work and the technical contribution may be the analysis under the stochastic setting. So could the authors summarize the contribution and challenges in deterministic setting?

Reference:

[1] Understanding Why Adam Outperforms SGD: Gradient Heterogeneity in Transformers.

[2] Understanding the Generalization of Adam in Learning Neural Networks with Proper Regularization.

**Questions:**

Please refer to __Weaknesses__.

---

### Note · Authors · 2025-11-30

I have read and agree with the venue's withdrawal policy on behalf of myself and my co-authors.